# ACTIVE-DORMANT ATTENTION HEADS: MECHANISTICALLY DEMYSTIFYING EXTREME-TOKEN PHENOMENA IN LLMS

## ABSTRACT

We investigate the mechanisms behind three puzzling phenomena observed in transformer-based large language models (LLMs): *attention sinks*, *value-state drains*, and *residual-state peaks*, collectively referred to the *extreme-token phenomena*. First, we demonstrate that these phenomena also arise in simpler architectures—transformers with one to three layers—trained on a toy model, the Bigram-Backcopy (BB) task. In this setting, we identify an *active-dormant mechanism* that causes attention heads to become attention sinks for certain domain-specific inputs while remaining non-sinks for others. We further develop a precise theoretical characterization of the training dynamics that lead to these phenomena, revealing that they are driven by a *mutual reinforcement mechanism*. By small interventions, we demonstrate ways to avoid extreme-token phenomena during pre-training. Next, we extend our analysis to pre-trained LLMs, including Llama and OLMo, revealing that many attention heads are governed by a similar active-dormant mechanism as in the BB task. We further show that the same mutual reinforcement mechanism drives the emergence of extreme-token phenomena during LLM pre-training. Our results study the mechanisms behind extreme-token phenomena in both synthetic and real settings and offer potential mitigation strategies.

## 1 INTRODUCTION

Recent analyses of transformer-based open-source large language models (LLMs), such as GPT-2 (Radford et al., 2019), Llama-2 (Touvron et al., 2023), Llama-3 (Dubey et al., 2024), Mixtral (Jiang et al., 2023), Pythia (Biderman et al., 2023), and OLMo (Groeneveld et al., 2024), have revealed several intriguing phenomena:

- **Attention sinks** (Xiao et al., 2023): In many attention heads, the initial token consistently attracts a large proportion of attention weights. In certain LLMs, other special tokens, such as the delimiter token, also draw significant attention. We refer to these as *sink tokens*.
- **Value state drains** (Guo et al., 2024): The value states of sink tokens are consistently much smaller than those of other tokens.
- **Residual state peaks** (Sun et al., 2024): The intermediate representations of sink tokens, excluding those from the first and last layers, exhibit a significantly larger norm than other tokens.

These phenomena often appear simultaneously, and we collectively refer to them as the **extreme-token phenomena**. Figure 1 illustrates these phenomena using a fixed prompt: "⟨s⟩ Summer is warm. Winter is cold." in Llama-3.1-8B-Base, where the first token, ⟨s⟩, the beginning-of-sentence token, serves as the sink token. We note that the first token does not have to be ⟨s⟩ to function as a sink token, as in GPT-2, where other tokens, being the initial token, can also serve this role. Furthermore, in models like Llama-2, a delimiter token can also act as the sink token. Despite the consistency of these observations, no prior work has provided a satisfying explanation for the mechanisms behind these phenomena. As a tentative explanation, Xiao et al. (2023) suggested that models tend to dump unnecessary attention values to specific tokens.

This work aims to demystify the extreme-token phenomena in LLMs. We show that the extreme-token phenomena are manifestations of the *active-dormant mechanism* of attention heads. We sup-

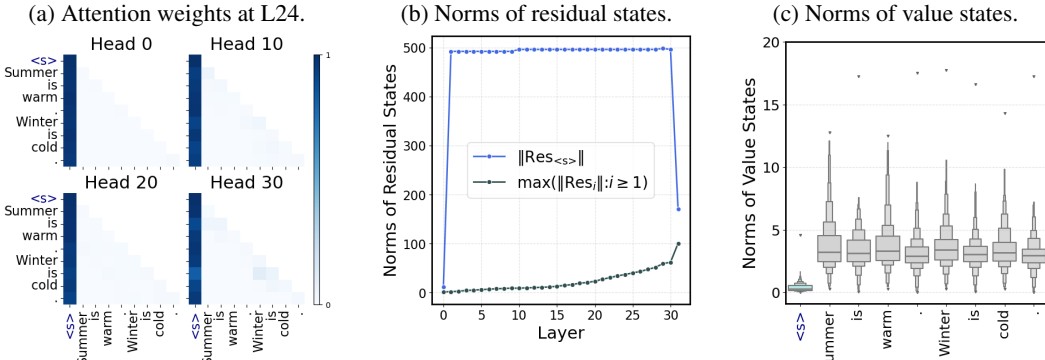

Figure 1: **Extreme-token phenomena in Llama 3.1-8B-Base.** We evaluate the sentence "⟨s⟩ Summer is warm. Winter is cold." on the Llama 3.1-8B-Base model. *Left (a):* The value of the attention weights across multiple heads at Layer 24. We demonstrate that there are *attention sinks*: the key state associated with the ⟨s⟩ token attracts the most attention from query states in these (and most) heads. *Middle (b):* The norm of the (residual stream) hidden states, measured at the output of each layer. We observe a *residual state peak* phenomenon: the ⟨s⟩ token's residual states have significantly larger norms than those of other tokens from layers 1 to 30. *Right (c):* The distribution of the norms of value states corresponding to each token at all layers and all heads. We observe the *value state drain* phenomenon: across many attention heads, the value state of the ⟨s⟩ token is much smaller than those of other tokens on average.

port this claim through studies on simplified transformer architectures and tasks, a dynamical theory of simplified models, and experiments on pre-trained LLMs. Our contributions are as follows:

1. In Section 2, we train one-to-three-layer transformers on the *Bigram-Backcopy* (BB) task, which also exhibits extreme-token phenomena similar to those observed in LLMs. We show that attention sinks and value-state drains are driven by the active-dormant mechanism mechanism. Both theoretically and empirically, we demonstrate that the mutual reinforcement dynamics underpin the extreme-token phenomena: attention sinks and value-state drains reinforce each other, leading to a stable phase where all query tokens produce identical attention logits for the keys of extreme tokens. Empirical evidence further shows that residual state peaks result from the interaction between this mutual reinforcement mechanism and Adam.

2. In Section 3, we demonstrate the *active-dormant mechanism* mechanism in LLMs by identifying an interpretable active-dormant head (Layer 16, Head 25 in Llama 2-7B-Base (Touvron et al., 2023)), confirmed through causal intervention analyses. We also discover circuits in LLMs related to extreme tokens that partially align with models trained on the BB task. Examining the dynamics of OLMo-7B-0424 (Groeneveld et al., 2024), we observe the same mutual reinforcement mechanism and stable phase, consistent with predictions from the BB task.

3. Through causal interventions, we isolate the extreme-token phenomena to architecture and optimization strategy. Specifically, we show that replacing SoftMax with ReLU activations in attention heads can eliminate extreme-token phenomena in the BB task, and switching from Adam to SGD removes the residual-state peak phenomenon in the BB task. Our work demonstrates potential classes of modifications to mitigate extreme-token phenomena in LLMs.

## 1.1 NOTATION

We denote the SoftMax attention layer with a causal mask as `attn`, the MLP layer as `mlp`, and the transformer block as TF. The query, key, value states, and residuals of a token $v$ are represented as $\mathrm{Qry}_v$, $\mathrm{Key}_v$, $\mathrm{Val}_v$, and $\mathrm{Res}_v$, respectively, with the specific layer and head indicated in context. We use ⟨s⟩ to refer to the "Beginning of Sequence" (bos) token. Throughout the paper, we employ zero-indexing (i.e., attention head and layer indices start from 0 rather than 1) for consistency between code and writing.

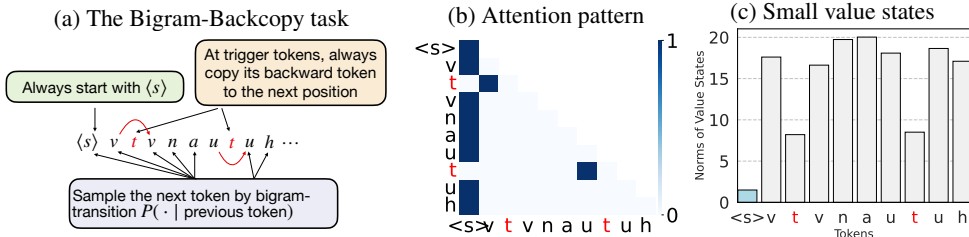

Figure 2: **Experiments on the Bigram-Backcopy task.** *Left (a):* We illustrate the data generation procedure for the Bigram-Backcopy task, where we fix 't', 'e', and the space character (' ') as trigger tokens. The BB task samples bigram transitions for non-trigger tokens and backcopies for trigger tokens. *Middle (b):* We present the attention weight heat map of a given prompt, with trigger tokens marked in red. Non-trigger tokens act as attention sinks. *Right (c):* We plot the value state norms for the prompt, where the $\langle s \rangle$ token has a tiny norm.

## 2  THE BIGRAM-BACKCOPY TASK

The Bigram-Backcopy task consists of two sub-tasks: *Bigram-transition* and *Backcopy*. Each input sequence begins with a $\langle s \rangle$ token, followed by tokens sampled according to a pre-determined bigram transition probability P. When some special trigger tokens are encountered, instead of sampling, the preceding token is copied to the next position. Following Bietti et al. (2024), we select the transition P and the vocabulary $\mathcal{V}$ with $|\mathcal{V}| = V = 64$ based on the estimated character-level bigram distribution from the tiny *Shakespeare* dataset. In all experiments, the set of trigger tokens $\mathcal{T}$ is fixed and consists of the $|\mathcal{T}| = 3$ most frequent tokens in the unigram distribution. Thus, the non-trigger token set, $\mathcal{V} \setminus \mathcal{T}$, comprises 61 tokens.

### 2.1  ONE-LAYER TRANSFORMER SHOWS ATTENTION SINKS AND VALUE-STATE DRAINS.

On the Bigram-Backcopy task, we pre-train a standard one-layer transformer with only one softmax `attn` head and one `mlp` layer. Unless otherwise specified, the model is trained with Adam for $10,000$ steps. We relegate the training details in Appendix C. Figure 2b shows that the trained transformer also exhibits the attention sink phenomenon, where the $\langle s \rangle$ token captures a significant proportion of the attention weights. More importantly, the attention weights reveal interpretable patterns: all non-trigger tokens exhibit attention sinks, while the attention for trigger tokens is concentrated on their preceding positions. Furthermore, Figure 2c reveals a value state drain phenomenon similar to LLMs, indicating that on non-trigger tokens, the `attn` head adds a minimal value to the residual stream.

**The active-dormant mechanism of the attention head:**  Inspired by the observed interpretable attention weight patterns, we propose the *active-dormant mechanism*. For any given token, an attention head is considered *active* if it contributes significantly to the residual state, and *dormant* if its contribution is minimal. As illustrated in Figure 2b, trained on the BB task, the attention head is active on trigger tokens and dormant on non-trigger tokens.

Figure 3a demonstrates that the `mlp` layer is responsible for the Bigram task whereas the `attn` head takes care of the Backcopy task. When the `mlp` layer is zeroed out, the backcopy loss remains significantly better than a random guess, but the bigram loss degrades to near-random levels. Conversely, when the `attn` layer is zeroed out, the backcopy loss becomes worse than a random guess, while the bigram loss remains unaffected. This suggests that on trigger tokens, the `attn` head is active and handles the backcopy task, whereas on non-trigger tokens, the `attn` head is dormant, allowing the `mlp` layer to handle the Bigram task. We summarize the active-dormant mechanism of the `attn` head in Claim 1.

**Claim 1.** *In the BB task, the* `attn` *head demonstrates active-dormant mechanism, alternating between two phases:*

- *Dormant phase: On non-trigger tokens, the* `attn` *head puts dominant weights to the* $\langle s \rangle$ *token, adding minimal value to the residual stream, having little impact on the model's output.*

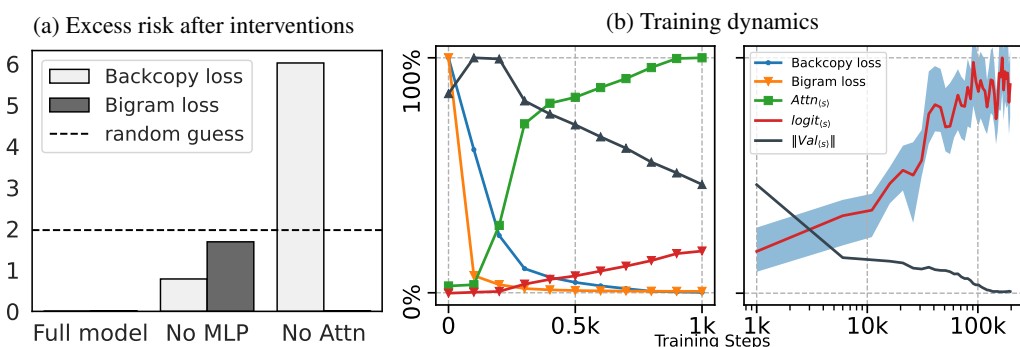

Figure 3: **Interventions and dynamics of one-layer transformer on the Bigram-Backcopy task.** *Left (a)*: We display the excess risks for a one-layer model trained on the Bigram-Backcopy (BB) task under various interventions. *Right (b)*: We plot the excess risks, attention weights, attention logits, and value state norms for the $\langle s \rangle$ token along the training dynamics. Each curve is rescaled to fall within a 0 to 1 range, though the trends remain consistent without rescaling. On the right side of *(b)*, the horizontal axis is logarithmically scaled. The $\text{logit}_{\langle s \rangle}$ curve denotes the mean of attention logits from all given non-trigger query tokens $v$ on the $\langle s \rangle$ token, normalized by the mean of attention logits on other tokens. The shaded area gives the 90% confidence interval on the distribution over all non-trigger tokens.

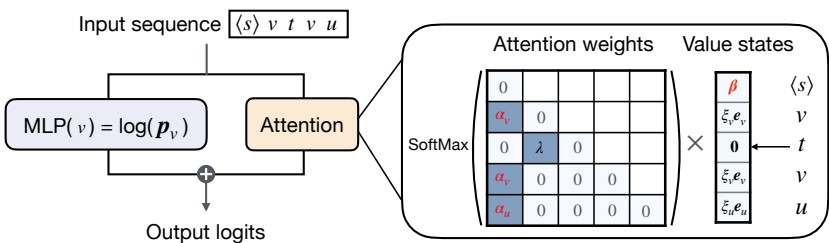

Figure 4: **The simplified transformer architecture with one `mlp`-layer and one `attn` head in parallel.** The predicted probability is the softmax of the output. Assume that the trainable variables are $(\boldsymbol{\alpha}, \boldsymbol{\beta}) \in \mathbb{R}^V \times \mathbb{R}^V$, which stands for the attention logits and value states of the $\langle s \rangle$ tokens.

- *Active phase: On trigger tokens, the `attn` head puts dominant weights to the relevant context tokens, adding substantial value states to the residual stream, resulting in a significant impact on the model's output.*

**The growth of attention logits on the $\langle \mathbf{s} \rangle$ token and the decrease in the norm of its value state.** Figure 3b displays the training dynamics of excess risks, attention weights, attention logits, and value state norms for the $\langle s \rangle$ token. All values are rescaled to highlight the trends. The backcopy excess risk and the bigram excess risk both drop to zero within the first 1000 steps. As the backcopy risk decreases, the attention weights on the $\langle s \rangle$ token increase, suggesting a relationship between the formation of attention sinks and the functional development of the attention heads. For each token $v_n$ at position $n$ in the prompt, we compute $\text{logit}_{\langle s \rangle} = \text{mean}_n[\langle \text{Qry}_{v_n}, \text{Key}_{\langle s \rangle} \rangle - \text{mean}_i(\langle \text{Qry}_{v_n}, \text{Key}_{v_i} \rangle)]$, which serves as a progress measure for attention sinks. Even after the attention weights on the $\langle s \rangle$ token is nearly 1, $\text{logit}_{\langle s \rangle}$ continues to increase. Simultaneously, the norm of the value state of the $\langle s \rangle$ token continues to decrease to a small value.

## 2.2 ANALYSIS OF A MINIMALLY-SUFFICIENT TRANSFORMER ARCHITECTURE

In this section, we analyze the training dynamics on the BB task by simplifying the architecture while preserving the attention sinks and value state drains phenomena. Let $\mathcal{V}$ denote the set of all tokens except the $\langle s \rangle$ token, and $\mathcal{T}$ denote the set of all trigger tokens. Given any $v \in \mathcal{V}$, we denote $p_{vk} = \text{P}(k|v)$ to be the next token Markov transition probability, and $\mathbf{p}_v = [p_{v1}, \ldots, p_{vV}]$ be the row vector in the simplex. We assume that the tokens are embedded into $V$-dimensional space

using one-hot encoding, and for notation simplicity, we abuse $v$ to stand for its one-hot encoding vector $\boldsymbol{e}_v \in \mathbb{R}^V$ which is a row vector. The predicted probability of the $n+1$ token is given by $\mathsf{SoftMax}(\mathrm{TF}([\langle s \rangle; v_{1:n-1}; v])_n)$, where transformer architecture is given by $\mathrm{TF}(\cdot) = \mathtt{attn}(\cdot) + \mathtt{mlp}(\cdot)$. Here $\mathtt{attn}(\cdot) = \mathsf{SoftMax}(\mathrm{mask}(\mathtt{Qry}(\cdot)\mathtt{Key}(\cdot)^\top))\mathtt{Val}(\cdot)$ and $(\mathtt{Qry}, \mathtt{Key}, \mathtt{Val})$ are linear maps from $\mathbb{R}^V \to \mathbb{R}^V$. Since the $\mathtt{mlp}$ layer could handle the Bigram task, we assume that $\mathtt{mlp}$ outputs the Markov transition probabilities $\mathbf{p}_v$ on non-trigger tokens $v$ and zero on trigger tokens. For the $\mathtt{attn}$ head, we assume that the attention logits on the $\langle s \rangle$ key-token are $(\alpha_{v_1}; \ldots; \alpha_{v_n})$, the attention logits on any trigger query-token are $(0, \ldots, \lambda, 0)$ where the second last coordinate is $\lambda$, and assume other logits are zero. Assume that the value state of $\langle s \rangle$ is $\boldsymbol{\beta} \in \mathbb{R}^V$, and the value state of each non-trigger token $v$ is a one-hot encoding vector $\boldsymbol{e}_v$ multiplied by $\xi_v \geq 0$. Figure 4 illustrates this simplified transformer architecture. These assumptions are summarized in the following equations.

$$
\begin{aligned}
\mathtt{mlp}(v) &= \log \mathbf{p}_v \cdot 1\{v \notin \mathcal{T}\} \quad \text{for } v \in \mathcal{V}, \\
\langle \mathtt{Qry}(v), \mathtt{Key}(\langle s \rangle) \rangle &= \alpha_v \cdot 1\{v \notin \mathcal{T}\} \quad \text{for } v \in \mathcal{V}, \\
\langle \mathtt{Qry}(v), \mathtt{Key}(v') \rangle &= \lambda \cdot 1\{v \in \mathcal{T}, v' \text{ is the former token of } v\} \quad \text{for } v, v' \in \mathcal{V}, \\
\mathtt{Val}(v) &= \xi_v \boldsymbol{e}_v \quad \text{with } \xi_v = 0 \text{ for } v \in \mathcal{T}, \text{ and } \xi_v \geq 0 \text{ for } v \in \mathcal{V} \setminus \mathcal{T}.
\end{aligned}
\tag{1}
$$

Theorem 2 demonstrates the existence of a transformer structure that is equivalent to the simplified version. We relegate the proof in Section B.

**Theorem 2.** *For any parameters $(\boldsymbol{\alpha} \in \mathbb{R}^V, \boldsymbol{\beta} \in \mathbb{R}^V, \boldsymbol{\xi} \in \mathbb{R}^V, \lambda \in \mathbb{R})$, there is a one-layer transformer $(\mathtt{mlp}, \mathtt{Qry}, \mathtt{Key}, \mathtt{Val})$ such that Eq. (1) holds. The transformer gives ground truth transition of the BB model if $\min_{v \in \mathcal{V}} \alpha_v \to \infty$, $\min_{v \in \mathcal{V}} \xi_v \to \infty$, $\lambda \to \infty$, and $\boldsymbol{\beta} = 0$.*

Throughout we adopt Eq. (1) as our assumption. We further define $W_k = \sum_{i=1}^n 1\{v_i = k\}$, $\boldsymbol{W} = (W_1, \ldots, W_V)$, and $W = \sum_{k \in \mathcal{V}} W_k = n$. Then for a non-trigger token $v$, the output of attention layer with input sequence $[\langle s \rangle; v_{1:n-1}; v]$ gives (denoting $\xi_k = 0$ for $k \in \mathcal{T}$)

$$
\mathrm{TF}([\langle s \rangle; v_{1:n-1}; v])_n = \log \mathbf{p}_v + \frac{e^{\alpha_v}}{e^{\alpha_v} + W} \boldsymbol{\beta} + \sum_{k=1}^V \frac{W_k \xi_k}{e^{\alpha_v} + W} \cdot \boldsymbol{e}_k.
$$

Therefore, on the non-trigger token $v$, the cross-entropy loss between the true Markov transition $\mathbf{p}_v$ and predicted transition $\mathsf{SoftMax}(\mathrm{TF}([v_{1:n-1}; v])_n)$ is given by

$$
\mathsf{loss}_v(\alpha_v, \boldsymbol{\beta}) = \sum_{k=1}^V p_{vk} \Big\{ \log \Big[ \sum_{i=1}^V p_{vi} \exp \Big( \frac{e^{\alpha_v} \beta_i + W_i \xi_i}{e^{\alpha_v} + W} \Big) \Big] - \frac{e^{\alpha_v} \beta_k + W_k \xi_k}{e^{\alpha_v} + W} - \log p_{vk} \Big\}.
$$

For simplicity, we neglect the loss on trigger tokens and assume that $(\{W_i\}_{i \in [V]}, W)$ are fixed across different positions in the input sequences[1], and consider the total loss to be the losses on each non-trigger token averaged with its proportion in the stable distribution $\{\pi_v\}_{v \in \mathcal{V}}$, given by

$$
\mathsf{loss}(\boldsymbol{\alpha}, \boldsymbol{\beta}) = \sum_{v \in \mathcal{V} \setminus \mathcal{T}} \pi_v \mathsf{loss}_v(\alpha_v, \boldsymbol{\beta}).
$$

**Theorem 3.** *Consider the gradient flow of the loss function $\mathsf{loss}(\boldsymbol{\alpha}, \boldsymbol{\beta})$. Assume $\xi_v \geq 0$ for any $v$, and $\{W_i \cdot \xi_i\}_{i \in \mathcal{V}}$ are not all equal.*

- *(Attention logits grow logarithmically reinforced by small value states) Fix $\boldsymbol{\beta} = \beta \cdot \mathbf{1}$ for a constant $\beta$, and consider the gradient flow over $\boldsymbol{\alpha}$. With any initial value $\boldsymbol{\alpha}(0)$, there exists $\boldsymbol{r}(t)$ with norm uniformly bounded in time such that*

$$
\boldsymbol{\alpha}(t) = \frac{1}{2} \log t \cdot \mathbf{1} + \boldsymbol{r}(t).
$$

- *(Value state shrinks to a small constant vector reinforced by large attention logits) Fix $\boldsymbol{\alpha} = \alpha \cdot \mathbf{1}$ for a constant $\alpha$, and define $\overline{\beta}(0) = V^{-1}[\sum_v \beta_v(0)]$. Consider the gradient flow over $\boldsymbol{\beta}$. As $t \to \infty$, we have*

$$
\boldsymbol{\beta}(t) \to \boldsymbol{\beta}^\star = \overline{\beta}(0) \cdot \mathbf{1} - e^{-\alpha} \cdot \boldsymbol{W} \circ \boldsymbol{\xi}.
$$

---

[1]We note that Reddy (2023) makes similar simplification in analyzing induction heads.

- *(Stable phase: identical attention logits) Consider the gradient flow over variables $(\boldsymbol{\alpha}, \boldsymbol{\beta})$. Any vector of the following form*

$$\boldsymbol{\alpha} = \alpha \cdot \mathbf{1}, \quad \boldsymbol{\beta} = c \cdot \mathbf{1} - e^{-\alpha} \cdot \boldsymbol{W} \circ \boldsymbol{\xi}, \quad \alpha, c \in \mathbb{R}$$

  *is a stationary point. These are all global minimizers of* $\mathsf{loss}(\boldsymbol{\alpha}, \boldsymbol{\beta})$.

The proof of Theorem 3 is provided in Appendix B.2. We give three key remarks: (1) As $\alpha_v \to \infty$, a Taylor expansion of the gradient $\partial \mathsf{loss}/\partial \alpha_v$ suggests that $\mathrm{d}\alpha_v/\mathrm{d}t \propto \exp(-2\alpha_v)$, which leads to the logarithmic growth of $\alpha_v$. Similar logarithmic growth exists in the literature under different setups (Tian et al., 2023a; Han et al., 2023). (2) For a fixed $\boldsymbol{\alpha} = \alpha\mathbf{1}$, under additional assumptions on the initial value $\boldsymbol{\beta}(0)$, we can prove a linear convergence for $\boldsymbol{\beta}$. (3) The stable phase described in Theorem 3 seems to imply that the system could be stable without attention sinks, as it does not require $\alpha$ to be large. However, in practice, models trained on the BB task tend to converge to a stable phase where $\alpha$ is relatively large.

**The Formation of Attention Sinks and Value State Drains.** When $\boldsymbol{\beta} = \mathbf{0}$, the attention logits on the $\langle \mathtt{s} \rangle$ token increase monotonically. This demonstrates that the presence of a small value state of the $\langle \mathtt{s} \rangle$ token reinforces the formation of attention sinks. When $\boldsymbol{\alpha} = \alpha \cdot \mathbf{1}$, with $\alpha$ sufficiently large, $\boldsymbol{\beta}(t) \to \overline{\beta}(0)\mathbf{1}$. Given the random Gaussian initialization, $\|\overline{\beta}(0)\mathbf{1}\|_2 \approx \|\boldsymbol{\beta}(0)\|_2/\sqrt{d}$, where $d$ is the hidden dimension. This demonstrates that the presence of attention sinks reinforces the formation of value states drains.

**Experimental verification.** Revisiting Figure 3b, which shows the dynamics of a full transformer model trained with Adam, we observe that both $\mathsf{logit}_{\langle \mathtt{s} \rangle}$ and $\|\mathtt{Val}_{\langle \mathtt{s} \rangle}\|_2$ exhibit growth rates consistent with Theorem 3. The $\mathsf{logit}_{\langle \mathtt{s} \rangle}$ is equivalent to $\alpha$ in this context, as all other attention logits are assumed to be zero under the setup of Theorem 3. When plotted on a logarithmic scale, the $\mathsf{logit}_{\langle \mathtt{s} \rangle}$ curve grows approximately linearly between 1,000 and 10,000 steps, then accelerates before stabilizing around 100,000 steps. Meanwhile, the norm of the value state decreases monotonically. The simultaneous increase in attention weights and decrease in value-state norms suggest that these phases occur together during the training process. To further validate Theorem 3, we construct a simplified model that aligns with Equ. (1), and train the parameters $(\boldsymbol{\alpha} \in \mathbb{R}^V, \boldsymbol{\beta} \in \mathbb{R}^V, \boldsymbol{\xi} \in \mathbb{R}^V, \lambda \in \mathbb{R})$ with Adam. The resulting training curves are similar to those of a one-layer transformer, also exhibiting the mutual reinforcement mechanism.

Combining theoretical insights and experimental evidence, we summarize the formation of attention sinks and value state drains as a mutual reinforcement mechanism.

**Claim 4** (Mutual reinforcement mechanism)**.** *For any attention head given a specific prompt, if the model can accurately predict the next token without the attention head, but adding any value state from previous tokens worsens the prediction, the attention head becomes dormant, forming an attention sink, leading to the mutual reinforcement of attention sinks and value state drains:*

1. *The SoftMax mechanism pushes the attention weights to the value state drains, reinforcing attention sinks.*

2. *The attention sinks on the value state drains further pushes down the value state, reinforcing value state drains.*

*The mutual reinforcement stabilizes at the phase when all tokens have identical large attention logits on the value state drains. Finally, due to the causal mask, the training dynamics favor the $\langle \mathtt{s} \rangle$ token to become an extreme token.*

We expect that the formation of extreme tokens in LLMs follows a similar mutual reinforcement mechanism. Indeed, although Theorem 3 focuses on a specific BB task with a simplified architecture and loss function, the same principles can be applied to more general scenarios. Specifically, for an attention head $\mathtt{attn}$, we assume that $(\mathtt{LLM} \setminus \mathtt{attn})(v) = \log \boldsymbol{p}_v$, meaning that the LLM, even if we zeroed out $\mathtt{attn}$, can still output an accurate next token prediction. Furthermore, we assume $\mathtt{Val}(v) = \xi_v \boldsymbol{e}_v$, indicating that adding the value state from any previous tokens performs a specific function. Under these assumptions, we expect the same theoretical results to apply to LLMs. In Section 3, we will explore the formation of attention sinks and value state drains along the training dynamics of LLMs, where we find empirical evidence that aligns with the theory.

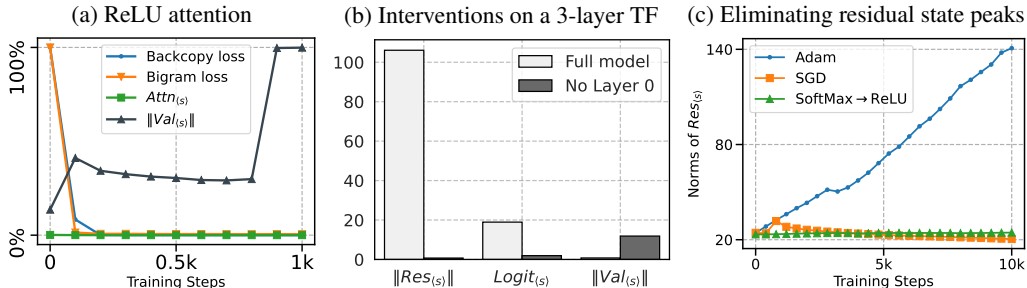

Figure 5: **Experiments on massive norms with multi-layer transformers trained on the Bigram-Backcopy task.** *Left (a):* We present the training dynamics of the ReLU attention for the first 1,000 steps. *Middle (b):* We plot the intervention results on the `attn`+`mlp`+`attn`+`mlp`+`mlp` structure. *Right (c):* We plot the evolution of massive norms in a three-layer transformer trained with Adam, SGD, and using a ReLU attention structure. Notably, only the three-layer model with softmax attention trained using Adam results in the emergence of residual state peaks.

**Replacing SoftMax by ReLU attention removes extreme-token phenomena.** As an implication of our theory, we predict that training with ReLU attention instead of SoftMax attention will eliminate the extreme-token phenomena. Without the SoftMax, the dynamics no longer push the attention weights on the $\langle s \rangle$ token, which remains zero along the training dynamics. Without attention sink, the dynamics no longer push down the value state norm, and the mutual reinforcement mechanism breaks. Figure 5a illustrates the training experiment on the BB task replacing SoftMax with ReLU, showing that both the Bigram and Backcopy risk match the Bayes risk after 200 training steps, but the attention logits of $\langle s \rangle$ do not grow, and the value state does not shrink, confirming the prediction.

### 2.3 THE EMERGENCE OF RESIDUAL STATE PEAKS

**The residual state peaks require a three-layer structure.** No residual state peaks appear in a one-layer transformer trained on the BB task. We train various models on the BB task and track the $\langle s \rangle$ token's residual state norms after layer 0. We relegate the experimental results to Appendix C. We find that a three-layer transformer is enough to produce residual state peaks. If we allow to skip some `mlp` or `attn` layers, the "`attn`+`mlp`+`attn`+`mlp`+`mlp`" combination becomes the simplest model that produces residual state peaks (Figure 10). Circuit analysis also reveals that LLMs typically add a large vector in the first layer and cancel it in the last layer. We propose that the add-then-cancel mechanism is essential for residual state peaks and requires at least three layers.

**Residual state peak reinforces attention sinks and value state drains in trained models.** Figure 5b presents the intervention results on the "`attn`+`mlp`+`attn`+`mlp`+`mlp`" model. We recenter the $\|\mathtt{Res}_{\langle s \rangle}\|_2$ by subtracting the average norm of other tokens from the $\langle s \rangle$ token norm. The $\mathrm{logit}_{\langle s \rangle}$ and $\|\mathtt{Val}_{\langle s \rangle}\|$ are computed in layer 1 following the same ways as in Figure 3b. When layer 0 is zeroed out, the residual norm returns to normal, attention logits decrease, and the value state norm rises. It verifies that the residual state peak contributes to the attention sink and value state drain phenomenon in the trained transformer.

**Replacing Adam by SGD removes the linear growth of residual state norm.** Figure 5c shows the $\langle s \rangle$'s residual state norms at the output of layer 0 of three-layer transformers with different configurations. Adam leads to a linear increase in residual norms. In contrast, with SGD, attention sinks persist, but residual state peaks vanish. The ReLU attention, which lacks the active-dormant mechanism, shows no residual state peaks.

## 3 EXTENDING PREDICTIONS OF THE BB MODEL TO LLMS

In this section, we examine extreme-token phenomena in open-source pre-trained LLMs. In Section 3.1, we analyze the static behavior of these phenomena in Llama 2-7B-Base (Touvron et al.,

2023), confirming that certain attention heads in LLMs exhibit both active and dormant phases. Notably, we identify a specific head that is active on GitHub samples but dormant on Wikipedia samples, illustrating the *active-dormant mechanism*. In Section 3.2, we explore the dynamic behavior of extreme-token phenomena during the pre-training process of OLMo-7B (Groeneveld et al., 2024). We show that the attention logits, value state norms, and residual state norms of the sink token(s) in OLMo mirror their behavior in the simpler BB model. Specifically, the simultaneous formation of attention sinks and value state drains gives evidence for the *mutual reinforcement mechanism*.

### 3.1 ACTIVE-DORMANT MECHANISM IN LLMS

Our study of the BB model leads to the following prediction about the extreme-token phenomena, which we hypothesize also applies to LLMs:

*Attention heads are controlled by an active-dormant mechanism. Attention sinks and value state drains indicate that an attention head is in dormant phase.*

This hypothesis suggests that in LLMs, attention heads become sinks or not depending on the context: the value vector can be totally non-informative towards picking likely next tokens for token distributions (e.g., tasks) in a particular context but not in others. This is a concrete instantiation vis-a-vis large-scale LLMs of the active-dormant dichotomy in Section 2, where this phenomenon was shown to occur in the context of small next-token predictors and the BB task.

Accordingly, we strive to find instances of heads in pretrained LLMs which satisfy this principle, i.e., which are dormant on some domains and active on others. In Figure 6, we show a particular attention head – Layer 16 Head 25 of Llama 2-7B-Base (Touvron et al., 2023) — which has an extremely clear active-dormant distinction across two distinct contexts (e.g., tokens from RedPajama (Computer, 2023) drawn from the GitHub subset versus the Wikipedia subset). While there are many such attention heads which are context-dependent — we provide some in Appendix D — we demonstrate this one because the conditions under which it is active are simple and interpretable, while others have more involved or complex criteria to become active. We observe that this attention head is *dormant* (i.e., an attention sink) on samples from Wikipedia, which more closely resemble prose, and *active* (i.e., not an attention sink) on samples from Github, which more closely resemble code. We also observe that this attention head, in general, contributes significantly to the performance of the model on code sequences, but has negligible impact on the performance of the model on prose sequences (Figure 6b). This is a further justification, from a practical perspective, of why this head is sometimes dormant and sometimes active — in some contexts we can ablate it from the model entirely with no effect, but in other contexts ablating the head leads to huge performance drops. We include more detail in Appendix E, where we extract a circuit for extreme-token phenomena in order to analyze the dormant-active mechanism and its interaction with the semantics of the input tokens.

### 3.2 TRAINING DYNAMICS OF EXTREME-TOKEN PHENOMENA IN LLMS

Our study of the BB model leads to the following prediction about the dynamical behavior of the extreme-token phenomena, which we hypothesize also applies to LLMs:

*The attention heads go through a attention-increasing and value-state-shrinking phase. They then go into a stable phase, with identical attention logits on the $\langle s \rangle$ token. Meanwhile, the residual state norm of the $\langle s \rangle$ token linearly increases during pre-training.*

We confirm these predictions below. To observe the training dynamics of a large-scale LLM, we use the setup of OLMo-7B-0424 (Groeneveld et al., 2024) (henceforth just referred to as OLMo), who have open-sourced weights at several steps during their training run. For our analysis, we inspect OLMo at a variety of training steps: every 500 steps throughout the first 10,000 steps, then 25,000 steps, then 50,000 steps, then every 50,000 steps until 449,000 steps (which is roughly the end of their training). Again, we use the input "Summer is warm. Winter is cold."[2] Notice that in this prompt, token 3, namely ".", is not very semantically meaningful; it becomes a sink token along with token 0 (c.f. Section 3.1, Appendix E, Appendix F.2).

---

[2] Note that OLMo does not have a $\langle s \rangle$ token, but attention sinks still form in the majority of heads. In particular, the first token behaves similarly to an attention sink. We discuss this in Appendix F.2.

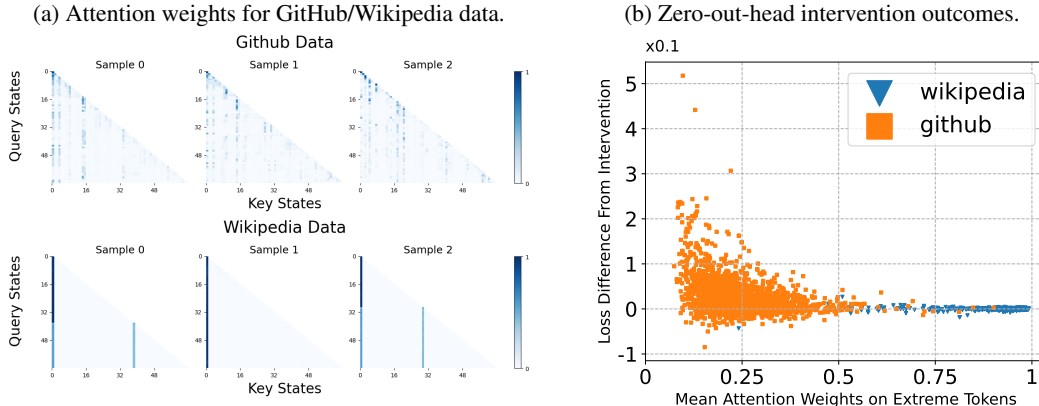

Figure 6: **Attention heads in LLMs are active on some domains and dormant on others.** For example, on Llama 2-7B-Base, we identify that Layer 16 Head 25 is active when the context contains many tokens related to programming, and dormant in other contexts such as prose. We use RedPajama-1T (Computer, 2023) Wikipedia and Github subsets for our data in this figure, truncating all samples to 64 tokens for demonstration purposes. *Left:* Sample weights from four randomly selected samples from each domain. *Right:* Result of an intervention study, i.e., change in cross-entropy of the input sequence when the attention head's output (concretely, the value states for this head) is manually set to zero, across sequences in both domains. We observe that the model's performance, measured by cross-entropy, strongly depends on the output of the attention head on coding data.

In Figure 7, we confirm that attention heads go through an attention-increasing and value-state-shrinking phase, and that the residual state norm of the $\langle s \rangle$ token increases linearly during pre-training. We show that, at Layer 24 of OLMo, the average attention on extreme tokens (token 0 and token 3) increases rapidly at the beginning of training and converges to a constant, while the value state norms of extreme tokens decrease rapidly. Also, the residual states of extreme tokens also increase linearly, while the rest quickly converge. In Figure 8 we show that attention heads converge to a stable phase, and that all logits corresponding to the first token's value states (i.e., all tokens' value of $\text{logit}_0$, except possibly the value of $\text{logit}_0$ corresponding to token 0 itself) have similar distributions. These confirm our dynamics insights from the BB model (c.f. Figure 3).

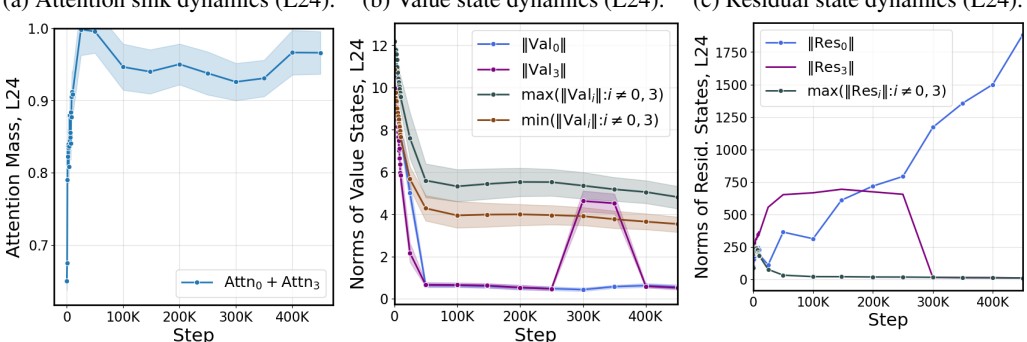

Figure 7: **Attention-increasing and value state-decreasing phase, and residual state norms.** *Left (a):* We plot the total attention mass on extreme tokens 0 and 3 at Layer 24 and averaged over all attention heads, during OLMo training. We observe that it increases rapidly and then maintains its value in $[0.9, 1]$ for the rest of training, which is in line with our predictions. *Middle (b):* We plot the norm of each token's value state at Layer 24 during training, averaged over all heads. We observe that the value states of all tokens shrink initially and then converge, while the value states of the extreme tokens shrink to much lower than all other tokens. *Right (c):* We plot the norm of each token's residual state at Layer 24 during training. We observe that the residual state of token 0 increases linearly in magnitude during training.

(a) Logit dynamics (L24).

(b) Logit statics (L24).

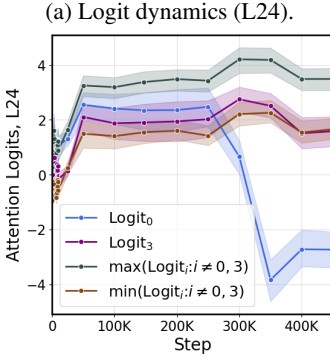
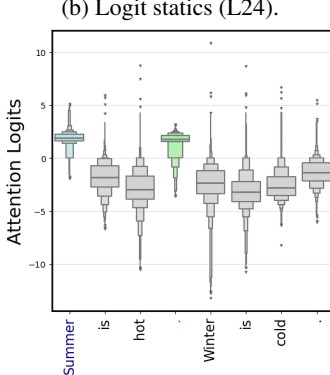

Figure 8: **Stable phase.** *Left (a):* We plot the normalized attention logits of all tokens' query states against token 0's key state during training. We observe that the logits of all non-extreme tokens' query states against token 0's key state in OLMo's Layer 24 are stable for a large fraction of the training run, after an initialization period. This echoes the stable phase prediction made in the BB model in Section 2. Note that this prediction makes no guarantees about the logit corresponding to the zeroth query token and zeroth key token, which will be set to 1 by the softmax and so its behavior is irrelevant for prediction. Also note that we use normalization, similar to Section 2, to make all terms comparable; namely we have $\text{logit}_i = \langle \texttt{Qry}_i, \texttt{Key}_0 \rangle - \text{mean}_j(\langle \texttt{Qry}_i, \texttt{Key}_j \rangle)$. *Right (b):* For this experiment, we generate 128 randomly sampled test tokens with IDs from 100 to 50000 in the OLMo tokenizer. We append each token separately to the test phrase "Summer is warm. Winter is cold.", creating 128 different samples, which we feed to the LLM to record the model behavior. We plot the distribution of (un-normalized) dot products $\langle \texttt{Qry}_{\text{test}}, \texttt{Key}_j \rangle$ across all heads at Layer 24 and all test tokens. We observe that logits of all regular tokens have very similar distributions, and the distributions of the logits corresponding to extreme tokens 0 and 3 are also similar. This confirms the hypothesis that at the end of training, attention heads converge to the stable phase, with similar logits on extreme tokens.

## 4 CONCLUSION

In this work, we investigated the *extreme-token phenomena*, namely *attention sinks*, *value state drains*, and *residual state peaks*. We analyzed a simple evocative model called the Bigram-Backcopy task, and theoretically and empirically showed that it exhibited the same extreme-token phenomena as in LLMs. Based on the Bigram-Backcopy task, we made several detailed predictions about the behavior of extreme-token phenomena in LLMs. In particular, we identified the *active-dormant mechanism* for attention heads in both the BB model and LLMs, of which attention sinks and value state drains are indicators, and a *mutual reinforcement mechanism* by which these phenomena are induced during pretraining. Using intuition about these mechanisms, we applied minor interventions to the model architecture and optimization procedure which disabled extreme-token phenomena within the BB model. Overall, our work uncovers the causes of extreme-token phenomena and points to possible pathways to eliminate them during LLM training.

We believe the most compelling direction for future work in this area is as follows. Specifically, one could build more performant and scalable interventions which would eliminate extreme-token phenomena and observe the effect on training dynamics and the finished model. This would make it easier to understand whether extreme token phenomena are necessary to build a powerful transformer-based LLM, whether they are merely helpful, or whether they are completely incidental to the particular architecture and optimization algorithms used by the community.

## ETHICS STATEMENT

This paper contributes towards the analysis of large language models. This paper does not add any ethical concerns beyond the usual ethics associated with use and analysis of large language models.

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

# A    RELATED WORKS

Several studies independently identified the "attention sink" phenomenon in language models and vision transformers, where attention weights were found to be concentrated on a few tokens (Xiao et al., 2023; Darcet et al., 2023; Han et al., 2023; Zhai et al., 2023; Elhage et al., 2023; Dettmers et al., 2022). Recent research has provided more detailed characterizations of this attention pattern and the attention sink phenomenon (Fu, 2024; Sun et al., 2024). Sun et al. (2024) attributed the attention sink to the massive activation of the hidden representations of the corresponding tokens. Both Sun et al. (2024) and Zhai et al. (2023) discussed methods for mitigating the attention sink by modifying the model and training recipes. Additionally, recent studies have leveraged the attention sink phenomenon to develop improved quantization and more efficient inference algorithms (Liu et al., 2024; Chen et al., 2024; Yu et al., 2024; Son et al., 2024).

The dynamics of transformers are studied under various simplifications, including linear attention structures (Zhang et al., 2023; Ahn et al., 2024), reparametrizations (Tian et al., 2023b), NTK (Deora et al., 2023), often in the setting of in-context linear regressions (Ahn et al., 2023; Wu et al., 2023; Zhang et al., 2024) and structured sequence (Bietti et al., 2024; Nichani et al., 2024; Tian et al., 2023a). Notably, Zhang et al. (2023) proves that a one-layer linear attention head trained with gradient descent converges to a model that implements the in-context linear regression algorithm. Huang et al. (2023); Kim et al. (2024) extend this to non-linear settings. Bietti et al. (2024) shows the fast learning of bigram memorization and the slow development of in-context abilities. Tian et al. (2023a) shows the scan and snap dynamics in reparametrized one-layer transformers. Reddy (2023) simplifies the structure of the induction head, showing the connection between the sharp transitions of in-context learning dynamics and the nested nonlinearities of multi-layer operations.

Mechanistic interpretability is a growing field focused on understanding the internal mechanisms of language models in solving specific tasks (Elhage et al., 2021; Geva et al., 2023; Meng et al., 2022; Nanda et al., 2023; Olsson et al., 2022; Bietti et al., 2024; Wang et al., 2022; Feng & Steinhardt, 2023; Todd et al., 2023). This includes mechanisms like the induction head and function vector for in-context learning (Elhage et al., 2021; Olsson et al., 2022; Todd et al., 2023; Bietti et al., 2024), the binding ID mechanism for binding tasks (Feng & Steinhardt, 2023), association-storage mechanisms for factual identification tasks (Meng et al., 2022), and a complete circuit for indirect object identification tasks (Wang et al., 2022). The task addressed in this paper is closely related to Bietti et al. (2024), which explored synthetic tasks where tokens are generated from either global or context-specific bigram distributions. Several other studies have also used synthetic tasks to investigate neural network mechanisms (Charton, 2022; Liu et al., 2022; Nanda et al., 2023; Allen-Zhu & Li, 2023; Zhu & Li, 2023; Guo et al., 2023; Zhang et al., 2022).

We note that Gurnee et al. (2024) proposed Attention Deactivation Neurons, a concept similar to Dormant Attention Heads. Gurnee et al. (2024) hypothesized that when such a head attends to the first token, it indicates that the head is deactivated and has minimal effect.

# B    PROOFS

Since we drop the trigger tokens in the loss function, we neglect $\mathcal{T}$ throughout the proof for notational convenience, assuming that $\mathcal{V}$ consists of only non-trigger tokens. We provide new notations which are frequently used in the proofs. Define the full bigram transition probability.

$$\mathbf{P} = \begin{pmatrix} p_{11} & \cdots & p_{1V} \\ \vdots & \ddots & \vdots \\ p_{V1} & \cdots & p_{VV} \end{pmatrix} = \begin{pmatrix} \boldsymbol{p}_1^\top \\ \vdots \\ \boldsymbol{p}_V^\top \end{pmatrix}. \tag{2}$$

Given token $v$, define the predicted probability, which is the logit output passed through the softmax activation

$$\boldsymbol{q}_v = \mathsf{SoftMax}(\mathrm{TF}([\langle \mathbf{s} \rangle; v_{1:n-1}; v])_n). \tag{3}$$

Similarly, define the full output probability matrix.

$$\mathbf{Q} = \begin{pmatrix} q_{11} & \cdots & q_{1V} \\ \vdots & \ddots & \vdots \\ q_{V1} & \cdots & q_{VV} \end{pmatrix} = \begin{pmatrix} \boldsymbol{q}_1^\top \\ \vdots \\ \boldsymbol{q}_V^\top \end{pmatrix}. \tag{4}$$

Given any vector $\boldsymbol{u} = [u_1; \ldots; u_d]$, define the corresponding diagonal matrix as

$$
\mathrm{diag}(\boldsymbol{u}) = \begin{pmatrix} u_1 & 0 & \ldots & 0 \\ \vdots & \ddots & & \vdots \\ \vdots & & \ddots & \vdots \\ 0 & \ldots & 0 & u_d \end{pmatrix}.
$$

Define

$$
\mathbf{G}_v^{\mathbf{Q}} = \mathrm{diag}(\boldsymbol{q}_v) - \boldsymbol{q}_v \boldsymbol{q}_v^\top \quad \mathbf{G}_v^{\mathbf{Q}} = \mathrm{diag}(\boldsymbol{p}_v) - \boldsymbol{p}_v \boldsymbol{p}_v^\top.
$$

Denote $\boldsymbol{z} = W \cdot \boldsymbol{\beta} - \boldsymbol{W} \circ \boldsymbol{\xi}$. We present a technical lemma.

**Lemma 5.** *The matrices $\mathbf{G}_v^{\mathbf{P}}$ and $\mathbf{G}_v^{\mathbf{Q}}$ are positive semi-definite for any $v$.*

*Proof.* Since we have that $\sum_{k=1}^V p_{vk} = 1$ and $\sum_{k=1}^V q_{vk} = 1$ for any $v$,

$$
(\mathbf{G}_v^{\mathbf{P}})_{ii} = p_i - p_i^2 = p_i(\sum_{k \neq i} p_k) \geq \sum_{k \neq i} |(\mathbf{G}_v^{\mathbf{P}})_{ik}|
$$

$$
(\mathbf{G}_v^{\mathbf{Q}})_{ii} = q_i - q_i^2 = q_i(\sum_{k \neq i} q_k) \geq \sum_{k \neq i} |(\mathbf{G}_v^{\mathbf{Q}})_{ik}|.
$$

This shows that both $\mathbf{G}_v^{\mathbf{P}}$ and $\mathbf{G}_v^{\mathbf{Q}}$ are diagonally dominant matrices. By Corollary 6.2.27 in Horn & Johnson (2012), they are positive semi-definite. $\qquad\square$

### B.1 PROOF OF THEOREM 2

We denote the hidden dimension as $d$ and the sequence length as $N$. We begin with the assumption regarding the transformer's positional embedding:

**Assumption A.** *For any token $v$ and position $i$, assume that the encoding combined with the positional embedding ensures that $\{\mathtt{ebd}(v_i)\}$ is linearly independent.*

Assumption A requires that $d \geq VN$. Given the fact that there are $O(\exp(d))$ approximately linearly independent vectors for large $d$ (Vershynin, 2018), it is possible to apply approximation theory to avoid Assumption A. However, since Assumption A pertains only to the construction of $\lambda$ for trigger tokens and is unrelated to Theorem 3, we adopt it to simplify the proof of Theorem 2.

*Proof.* Consider vectors $\mathbf{u}_i \in \mathbb{R}^d$, $i \in [N]$ such that $\mathbf{u}_i^\top \mathbf{u}_j = 0$, $i \neq j$, and $\mathbf{u}_i^\top \mathtt{ebd}(v_j)$ for any $v \in \mathcal{V}$ and $i, j \in [N]$. Adopting Assumption A, there exists a matrix $\mathtt{Qry}$ such that

$$
\begin{aligned}
\mathtt{Qry}(\mathtt{ebd}(v_i)) &= \lambda \mathbf{u}_{i-1} \quad \text{for } v_i \in \mathcal{T}, \ i > 1, \\
\mathtt{Qry}(\mathtt{ebd}(v_i)) &= \alpha_{v_i} \mathbf{u}_0 \quad \text{for } v_i \in \mathcal{V} \setminus \mathcal{T}, \ i > 0.
\end{aligned}
\tag{5}
$$

Define the corresponding key matrix.

$$
\begin{aligned}
\mathtt{Key}(\mathtt{ebd}(v_i)) &= \mathbf{u}_i \quad \text{for } v_i \in \mathcal{V}, \ i > 0, \\
\mathtt{Key}(\mathtt{ebd}(\langle \mathtt{s} \rangle)) &= \mathbf{u}_0.
\end{aligned}
\tag{6}
$$

There exists a value matrix $\mathtt{Val}$ such that

$$
\begin{aligned}
\mathtt{Val}(\mathtt{ebd}(v_i)) &= 0 \quad \text{for } v_i \in \mathcal{T}, \ i > 1, \\
\mathtt{Val}(\mathtt{ebd}(v_i)) &= \xi_{v_i} \mathbf{u}_i \quad \text{for } v_i \in \mathcal{V} \setminus \mathcal{T}, \ i > 0, \\
\mathtt{Val}(\mathtt{ebd}(\langle \mathtt{s} \rangle)) &= \boldsymbol{\beta}.
\end{aligned}
\tag{7}
$$

Further define the matrix $\mathbf{M}$ that satisfies

$$
\begin{aligned}
\mathbf{M}(\mathtt{ebd}(v_i)) &= \log \mathbf{p}_{v_i} \cdot 1\{v_i \notin \mathcal{T}\} \quad \text{for } v_i \in \mathcal{V}, \ i \in [N], \\
\mathbf{M}(\mathbf{u}_i) &= \boldsymbol{e}_i \quad \text{for } i \in [N].
\end{aligned}
\tag{8}
$$

Setting $\mathtt{mlp}(\cdot) = \mathrm{ReLU}(\mathbf{M}(\cdot))$, we can then verify that the residual connection gives that $\mathrm{TF}([\langle \mathtt{s} \rangle; v_{1:n-1}; v_n]) = \mathtt{mlp}(\mathtt{ebd}(v_n) + \mathtt{attn}(\mathtt{ebd}(v_n)))$, which is equivalent to the simplified model.

When $\min_{v \in \mathcal{V}} \alpha_v \to \infty$, $\min_{v \in \mathcal{V}} \xi_v \to \infty$, $\lambda \to \infty$, and $\boldsymbol{\beta} = 0$, if $v_n \in \mathcal{T}$, $\mathsf{SoftMax}[\mathrm{TF}([\langle \mathtt{s} \rangle; v_{1:n-1}; v_n])] = \delta_{v_{n-1}}$. If $v_n \in \mathcal{V} \setminus \mathcal{T}$, $\mathsf{SoftMax}[\mathrm{TF}([\langle \mathtt{s} \rangle; v_{1:n-1}; v_n])] = \boldsymbol{p}_{v_n}$. All next-token probabilities match those in the data-generating procedure, aligning with the oracle algorithm. $\qquad\square$

## B.2   THE STABLE PHASE IN THEOREM 3

Lemma 6 computes the gradient of $\mathbf{Q}$.

**Lemma 6.** *We have*

$$\frac{\partial q_{ik}}{\partial \alpha_v} = \frac{\mathbf{1}\{i=v\}q_{ik}e^{\alpha_i}}{(e^{\alpha_i}+W)^2}\Big[W\beta_k - W_k\xi_k - \sum_{j=1}^{V}q_{ij}(W\beta_j - W_j\xi_j)\Big],$$

$$\frac{\partial q_{ik}}{\partial \beta_v} = \frac{e^{\alpha_i}}{e^{\alpha_i}+W}[q_{ik}\mathbf{1}\{k=v\} - q_{ik}q_{iv}].$$

*Furthermore,*

$$\sum_{v=1}^{V}\frac{\partial q_{ik}}{\partial \alpha_v} = 0, \quad \sum_{v=1}^{V}\frac{\partial q_{ik}}{\partial \beta_v} = 0.$$

*Proof.* We repeatedly use the following two facts:

$$\frac{\partial\Big\{\exp\Big[\frac{W_k\xi_k+e^{\alpha_i}\beta_k}{e^{\alpha_i}+W}\Big]\Big\}}{\partial \alpha_v} = \frac{e^{\alpha_v}(W\alpha_k - W_k\xi_k)}{(e^{\alpha_i}+W)^2}\exp\Big[\frac{W_k\xi_k+e^{\alpha_i}\beta_k}{e^{\alpha_i}+W}\Big],$$

$$\frac{\partial\Big\{\exp\Big[\frac{W_k\xi_k+e^{\alpha_i}\beta_k}{e^{\alpha_i}+W}\Big]\Big\}}{\partial \beta_v} = \frac{\mathbf{1}\{i=v\}e^{\alpha_i}}{e^{\alpha_i}+W}\exp\Big[\frac{W_k\xi_k+e^{\alpha_i}\beta_k}{e^{\alpha_i}+W}\Big].$$

When $i \neq v$, $q_{ik}$ does not include $\alpha_v$, making the gradients as zero. When $i = v$, we have

$$\frac{\partial q_{vk}}{\partial \alpha_v} = q_{vk}e^{\alpha_v}\Big[\frac{W\beta_k - W_k\xi_k}{(e^{\alpha_v}+W)^2}\Big] - \frac{q_{vk}\sum_{i=1}^{V}p_{vi}e^{\alpha_v}\Big[\frac{W\beta_i - W_i\xi_i}{(e^{\alpha_v}+W)^2}\Big]\exp\Big[\frac{W_i\xi_i+e^{\alpha_v}\beta_i}{e^{\alpha_v}+W}\Big]}{\sum_{i=1}^{V}p_{vi}\exp\Big[\frac{W_i\xi_i+e^{\alpha_v}\beta_i}{e^{\alpha_v}+W}\Big]}$$

$$= \frac{e^{\alpha_v}}{(e^{\alpha_v}+W)^2}\Big\{q_{vk}[W\beta_k - W_k\xi_k] - q_{vk}\sum_{j=1}^{V}q_{vj}^{\top}(W\alpha_j - W_j\xi_j)\Big\},$$

and

$$\frac{\partial q_{ik}}{\partial \beta_v} = \Big[\frac{e^{\alpha_i}}{e^{\alpha_i}+W}\Big]q_{ik}\mathbf{1}\{k=v\} - \frac{\Big[\frac{e^{\alpha_i}}{e^{\alpha_i}+W}\Big]p_{iv}\exp\Big[\frac{W_v\xi_v+e^{\alpha_i}\beta_v}{e^{\alpha_i}+W}\Big]p_{iv}\exp\Big[\frac{W_k\xi_k+e^{\alpha_i}\beta_k}{e^{\alpha_i}+W}\Big]}{\Big(\sum_{j=1}^{V}p_{jv}j\exp\Big[\frac{W_j\xi_j+e^{\alpha_i}\beta_j}{e^{\alpha_i}+W}\Big]\Big)^2}$$

$$= \Big[\frac{e^{\alpha_i}}{e^{\alpha_i}+W}\Big][q_{ik}\mathbf{1}\{k=v\} - q_{ik}q_{iv}].$$

We can verify that

$$\sum_{v=1}^{V}\frac{\partial q_{ik}}{\partial \alpha_v} = \frac{e^{\alpha_v}}{(e^{\alpha_v}+W)^2}\sum_{v=1}^{V}\Big\{q_{vk}[W\beta_k - W_k\xi_k] - q_{vk}\sum_{j=1}^{V}q_{vj}^{\top}(W\alpha_j - W_j\xi_j)\Big\}$$

$$= \frac{e^{\alpha_v}}{(e^{\alpha_v}+W)^2}\Big\{\sum_{v=1}^{V}q_{vk}[W\beta_k - W_k\xi_k] - \sum_{j=1}^{V}q_{vj}^{\top}(W\alpha_j - W_j\xi_j)\Big\}$$

$$= 0,$$

and

$$\sum_{v=1}^{V}\frac{\partial q_{ik}}{\partial \beta_v} = \Big[\frac{e^{\alpha_i}}{e^{\alpha_i}+W}\Big]\sum_{v=1}^{V}[q_{ik}\mathbf{1}\{k=v\} - q_{ik}q_{iv}]$$

$$= \Big[\frac{e^{\alpha_i}}{e^{\alpha_i}+W}\Big][q_{iv} - q_{iv}]$$

$$= 0.$$

This finishes the proof of Lemma 6. $\qquad\square$

Proposition 7 computes the gradient of loss with respect to $\boldsymbol{\alpha}$ and $\boldsymbol{\beta}$, giving the gradient flow.

**Proposition 7.** *The gradient flow of optimizing* $\text{loss}(\boldsymbol{\alpha}, \boldsymbol{\beta})$ *is given by*

$$\dot{\alpha}_v(t) = \frac{\pi_v e^{\alpha_v}}{(e^{\alpha_v} + W)^2} \sum_{i=1}^{V} (p_{vi} - q_{vi})(W\beta_i - W_i\xi_i),$$

$$\dot{\beta}_v(t) = \sum_{k=1}^{V} \left\{ \frac{\pi_k e^{\alpha_k}[p_{kv} - q_{kv}]}{e^{\alpha_k + W}} \right\}.$$

*Proof.* The gradient flow gives that

$$\dot{\alpha}_v(t) = -\frac{\partial \text{loss}(\boldsymbol{\alpha}, \boldsymbol{\beta})}{\partial \alpha_v}, \quad \text{and} \quad \dot{\beta}_v(t) = -\frac{\partial \text{loss}(\boldsymbol{\alpha}, \boldsymbol{\beta})}{\partial \beta_v}.$$

Taking the derivative of $\text{loss}(\boldsymbol{\alpha}, \boldsymbol{\beta})$ gives that

$$\frac{\partial \text{loss}(\boldsymbol{\alpha}, \boldsymbol{\beta})}{\partial \alpha_v} = \pi_v \sum_{k=1}^{V} p_{vk} \cdot \frac{-1}{q_{vi}} \cdot \frac{\partial q_{vi}}{\partial \alpha_v}$$

$$= \frac{\pi_v e^{\alpha_v}}{(e^{\alpha_v} + W)^2} \left\{ \sum_{i=1}^{V} q_{vi}[W\beta_i - W_i\xi_i] - \sum_{k=1}^{V} p_{vk}[W\beta_k - W_k\xi_k] \right\}$$

$$= \frac{\pi_v e^{\alpha_v}}{(e^{\alpha_v} + W)^2} \sum_{k=1}^{V} \left\{ [q_{vk} - p_{vk}][W\beta_k - W_k\xi_k] \right\}.$$

Similarly, we have that

$$\frac{\partial \text{loss}(\boldsymbol{\alpha}, \boldsymbol{\beta})}{\partial \beta_v} = \sum_{j=1}^{V} \pi_j \sum_{k=1}^{V} p_{jk} \left\{ \frac{e^{\alpha_j} q_{jv}}{e^{\alpha_j} + W} - \frac{e^{\alpha_j} \mathbf{1}\{k = v\}}{e^{\alpha_j} + W} \right\}$$

$$= \sum_{j=1}^{V} \left\{ \frac{\pi_j e^{\alpha_j}[q_{jv} - p_{jv}]}{e^{\alpha_j} + W} \right\}.$$

This proves Proposition 7. $\qquad\square$

**Theorem 8** (Restatement the stable phase part in Theorem 3)**.** *Consider the gradient flow of optimizing* $\text{loss}(\boldsymbol{\alpha}, \boldsymbol{\beta})$*. The gradient flow has sink stationary points*

$$\boldsymbol{\alpha}^\star = \alpha \mathbf{1}, \quad \boldsymbol{\beta}^\star = c \cdot \mathbf{1} - e^{-\alpha} \cdot \boldsymbol{W} \circ \boldsymbol{\xi}.$$

*Proof.* When $\boldsymbol{\alpha} = \boldsymbol{\alpha}^\star$ and $\boldsymbol{\beta} = \boldsymbol{\beta}^\star$,

$$q_{vi} = \frac{p_{vi} \exp\left[\frac{W_i\xi_i + e^\alpha \beta_i}{e^\alpha + W}\right]}{\sum_{k=1}^{V} p_{vk} \exp\left[\frac{W_k\xi_k + e^\alpha \beta_k}{e^\alpha + W}\right]}$$

$$= \frac{p_{vi} \exp\left[\frac{c}{e^\alpha + W}\right]}{\sum_{k=1}^{V} p_{vk} \exp\left[\frac{c}{e^\alpha + W}\right]}$$

$$= p_{vi}.$$

Take $q_{vi}$'s into $\partial \text{loss}(\boldsymbol{\alpha}, \boldsymbol{\beta})/\partial\boldsymbol{\alpha}$ and $\partial \text{loss}(\boldsymbol{\alpha}, \boldsymbol{\beta})/\partial\boldsymbol{\beta}$.

$$\frac{\partial \text{loss}(\boldsymbol{\alpha}, \boldsymbol{\beta})}{\partial \alpha_v}\bigg|_{\boldsymbol{\alpha}^\star, \boldsymbol{\beta}^\star} = \frac{\pi_v e^{\alpha_v}}{(e^{\alpha_v} + W)^2} \sum_{k=1}^{V} \left\{ (q_{vk} - p_{vk})[W\beta_k - W_k\xi_k] \right\} = 0,$$

$$\frac{\partial \text{loss}(\boldsymbol{\alpha}, \boldsymbol{\beta})}{\partial \beta_v}\bigg|_{\boldsymbol{\alpha}^\star, \boldsymbol{\beta}^\star} = \sum_{k=1}^{V} \left\{ \frac{\pi_k e^{\alpha_k}[q_{kv} - p_{kv}]}{e^{\alpha_k} + W} \right\} = 0.$$

This shows that the given points are stationary points. We further compute the second-order derivative using Lemma 6.

$$\frac{\partial^2 \text{loss}(\boldsymbol{\alpha}, \boldsymbol{\beta})}{\partial \alpha_i \partial \alpha_v}\bigg|_{\boldsymbol{\alpha}^\star, \boldsymbol{\beta}^\star} = \mathbf{1}\{v = i\} \cdot \frac{\pi_v e^\alpha}{(e^\alpha + W)^2} \sum_{k=1}^{V} \left\{ \frac{\partial q_{ik}}{\partial \alpha_v} [W\beta_k - W_k \xi_k] \right\}$$

$$= \mathbf{1}\{v = i\} \cdot \frac{-\pi_v e^{2\alpha}}{(e^\alpha + W)^4} \left\{ \sum_{k=1}^{V} q_{ik}(e^{-\alpha}W + W_k)^2 \xi_k^2 - \left[ \sum_{k=1}^{V} q_{ik}(e^{-\alpha}W + W_k)\xi_k \right]^2 \right\},$$

$$= \mathbf{1}\{v = i\} \cdot \frac{-\pi_v e^{2\alpha}}{(e^\alpha + W)^4} \left\{ \sum_{k=1}^{V} p_{ik}(e^{-\alpha}W + W_k)^2 \xi_k^2 - \left[ \sum_{k=1}^{V} p_{ik}(e^{-\alpha}W + W_k)\xi_k \right]^2 \right\}.$$

where in the second line, we take $\beta_k^\star = c - e^{-\alpha}\xi_k$ and use that $\sum_{k=1}^{V} \partial q_{ik}/\partial \alpha_v = 0$. In the last line, we take $\mathbf{Q} = \mathbf{P}$. Similarly, we compute the gradients with respect to $\alpha_i$ and $\beta_v$.

$$\frac{\partial^2 \text{loss}(\boldsymbol{\alpha}, \boldsymbol{\beta})}{\partial \alpha_i \partial \beta_v}\bigg|_{\boldsymbol{\alpha}^\star, \boldsymbol{\beta}^\star} = \frac{\pi_i e^\alpha}{(e^\alpha + W)^2} \sum_{k=1}^{V} \left\{ \frac{\partial q_{ik}}{\partial \beta_v} [W\beta_k - W_k \xi_k] \right\}$$

$$= \frac{p_{iv} \pi_i e^{2\alpha}}{(e^\alpha + W)^3} \left\{ -(e^{-\alpha}W + W_k)\xi_k + \sum_{k=1}^{V} p_{ik}(e^{-\alpha}W + W_k)\xi_k \right\}.$$

With the same manner, we compute the gradients with respect to $\beta_i$ and $\beta_v$.

$$\frac{\partial^2 \text{loss}(\boldsymbol{\alpha}, \boldsymbol{\beta})}{\partial \beta_i \partial \beta_v}\bigg|_{\boldsymbol{\alpha}^\star, \boldsymbol{\beta}^\star} = \sum_{k=1}^{V} \left\{ \frac{\partial q_{ki}}{\partial \beta_v} \frac{\pi_k e^\alpha}{e^\alpha + W} \right\}$$

$$= \frac{e^{2\alpha}}{(e^\alpha + W)^2} \sum_{k=1}^{V} [\mathbf{1}\{v = i\} p_{kv} - p_{ki} p_{kv}].$$

Define $\mathbf{z} = [z_1; \ldots; z_V]$ so that $z_k = -(e^{-\alpha}W + W_k)\xi_k$. Combining above computations gives that

$$\text{Hessian}(\text{loss}(\boldsymbol{\alpha}^\star, \boldsymbol{\beta}^\star)) = \begin{pmatrix} \nabla_{\boldsymbol{\alpha}}^2 \text{loss}(\boldsymbol{\alpha}, \boldsymbol{\beta}) & \nabla_{\boldsymbol{\alpha}} \nabla_{\boldsymbol{\beta}} \text{loss}(\boldsymbol{\alpha}, \boldsymbol{\beta}) \\ \nabla_{\boldsymbol{\beta}} \nabla_{\boldsymbol{\alpha}} \text{loss}(\boldsymbol{\alpha}, \boldsymbol{\beta}) & \nabla_{\boldsymbol{\alpha}}^2 \text{loss}(\boldsymbol{\alpha}, \boldsymbol{\beta}) \end{pmatrix},$$

with

$$\nabla_{\boldsymbol{\alpha}}^2 \text{loss}(\boldsymbol{\alpha}, \boldsymbol{\beta}) = \frac{e^{2\alpha}}{(e^\alpha + W)^4} \text{diag} \left\{ \boldsymbol{\pi} \circ [\mathbf{z}^\top \mathbf{G}_1^{\mathbf{P}} \mathbf{z}; \ldots; \mathbf{G}_V^{\mathbf{P}} \mathbf{z}] \right\},$$

$$\nabla_{\boldsymbol{\alpha}} \nabla_{\boldsymbol{\beta}} \text{loss}(\boldsymbol{\alpha}, \boldsymbol{\beta}) = \frac{e^{2\alpha}}{(e^\alpha + W)^3} \text{diag} \left\{ \boldsymbol{\pi} \right\} [\mathbf{z}^\top \mathbf{G}_1^{\mathbf{P}}; \ldots; \mathbf{z}^\top \mathbf{G}_V^{\mathbf{P}}],$$

$$\nabla_{\boldsymbol{\beta}}^2 \text{loss}(\boldsymbol{\alpha}, \boldsymbol{\beta}) = \frac{e^{2\alpha}}{(e^\alpha + W)^2} \sum_{k=1}^{V} \pi_k \mathbf{G}_k^{\mathbf{P}}.$$

At last, we diagonalize the Hessian matrix and get that

$$\text{Diag-Hessian}(\text{loss}(\boldsymbol{\alpha}^\star, \boldsymbol{\beta}^\star)) = \begin{pmatrix} \nabla_{\boldsymbol{\alpha}}^2 \text{loss}(\boldsymbol{\alpha}, \boldsymbol{\beta}) & 0 \\ 0 & \frac{e^{2\alpha}}{(e^\alpha + W)^2} \mathbf{H} \end{pmatrix},$$

where the $\mathbf{H}$ is given by

$$\mathbf{H} = \sum_{k=1}^{V} \pi_k \left( \mathbf{G}_k^{\mathbf{P}} - (\boldsymbol{z}^\top \mathbf{G}_k^{\mathbf{P}} \boldsymbol{z})^{-1} \mathbf{G}_k^{\mathbf{P}} \boldsymbol{z} \boldsymbol{z}^\top \mathbf{G}_k^{\mathbf{P}} \right).$$

To prove that $\mathbf{H}$ is positive semi-definite, consider any vector $\boldsymbol{\eta}$ with $\|\boldsymbol{\eta}\|_2 = 1$.

$$\boldsymbol{\eta}^\top \mathbf{H} \boldsymbol{\eta} = \sum_{k=1}^{V} \pi_k \left( \boldsymbol{\eta}^\top \mathbf{G}_k^{\mathbf{P}} \boldsymbol{\eta} - \frac{\boldsymbol{\eta}^\top \mathbf{G}_k^{\mathbf{P}} \boldsymbol{z} \boldsymbol{z}^\top \mathbf{G}_k^{\mathbf{P}} \boldsymbol{\eta}}{\boldsymbol{z}^\top \mathbf{G}_k^{\mathbf{P}} \boldsymbol{z}} \right).$$

Since $\mathbf{G}_k^{\mathbf{P}}$'s are positive semi-definite, the Cauchy inequality gives that

$$\boldsymbol{z}^\top \mathbf{G}_k^{\mathbf{P}} \boldsymbol{\eta} \leq \sqrt{\boldsymbol{z}^\top \mathbf{G}_k^{\mathbf{P}} \boldsymbol{z} \boldsymbol{\eta}^\top \mathbf{G}_k^{\mathbf{P}} \boldsymbol{\eta}}.$$

As a result, we have that

$$\boldsymbol{\eta}^\top \mathbf{H} \boldsymbol{\eta} \geq \sum_{k=1}^{V} \pi_k \left( \boldsymbol{\eta}^\top \mathbf{G}_k^{\mathbf{P}} \boldsymbol{\eta} - \frac{\boldsymbol{z}^\top \mathbf{G}_k^{\mathbf{P}} \boldsymbol{z} \boldsymbol{\eta}^\top \mathbf{G}_k^{\mathbf{P}} \boldsymbol{\eta}}{\boldsymbol{z}^\top \mathbf{G}_k^{\mathbf{P}} \boldsymbol{z}} \right) = 0.$$

This shows that $\mathbf{H}$ is positive semi-definte. Therefore, $\mathrm{Hessian}(\mathsf{loss}(\boldsymbol{\alpha}^\star, \boldsymbol{\beta}^\star))$ is positive semi-definte. This proves Theorem 8. $\qquad\square$

We prove Theorem 8 through direct computation. Due to the non-linearity, it's unclear whether other stationary points exist. However, we observe that all of our simulations converge to the given stationary points.

### B.3 ATTENTION SINKS IN THEOREM 3

**Theorem 9** (Restatement of the attention sink part in Theorem 3). *Fixing $\boldsymbol{\beta} = c \cdot \mathbf{1}$, with any initial value, there exists $\boldsymbol{r}(t)$ with bounded norm such that*

$$\boldsymbol{\alpha}(t) = \frac{1}{2} \log t \cdot \mathbf{1} + \boldsymbol{r}(t).$$

*Proof.* We separately analyze each entry of $\boldsymbol{\alpha}$. Focusing on $\alpha_v$, to simplify the notation, we introduce a random variable $\varphi$ such that $\mathbb{P}(\varphi = W_k \xi_k) = p_{vk}$. Define

$$u = e^{\alpha_v}.$$

Therefore, using Lemma 7, we get that

$$\frac{\mathrm{d}u}{\mathrm{d}t} = \frac{\pi_v e^{2\alpha_v}}{(e^{\alpha_v} + W)^2} \sum_{i=1}^{V} (q_{vi} - p_{vi})(W\beta_i - W_i \xi_i).$$

We take in $\beta = c$ and expand the expression of $\mathrm{d}u/\mathrm{d}t$. This gives us

$$\frac{\mathrm{d}u}{\mathrm{d}t} = \frac{\pi_v u^2}{(u+W)^2} \frac{\sum_{k=1}^{V} p_{vk} e^{W_k \xi_k/(u+W)} W_k \xi_k - \sum_{k=1}^{V} p_{vk} e^{W_k \xi_k/(u+W)} \sum_{k=1}^{V} W_k \xi_k}{\sum_{k=1}^{V} p_{vk} e^{W_k \xi_k/(u+W)}}$$

$$= \frac{\pi_v u^2}{(u+W)^2} \frac{\mathrm{Cov}(e^{\frac{\varphi}{u+W}}, \varphi)}{\mathbb{E} e^{\frac{\varphi}{u+W}}}.$$

Since both $e^{x/(u+W)}$ and $x$ are monotonically increasing with respect to $x$, $u$ is monotonically increasing. This means that

$$\frac{u(t)^2}{[u(t)+W]^2} \geq \frac{u(0)^2}{[u(0)+W]^2}, \quad \mathbb{E} e^{\frac{\varphi}{u(t)+W}} \leq \mathbb{E} e^{\frac{\varphi}{u(0)+W}}.$$

Meanwhile, if we consider the first and second order approximation of $e^{\varphi/(u+W)}$,

$$e^{\frac{\varphi}{u+W}} = 1 + \frac{\theta_1(\varphi)\varphi}{u+W}, \quad e^{\frac{\varphi}{u+W}} = 1 + \frac{\varphi}{u+W} + \theta_2(\varphi)\left[\frac{\varphi}{u+W}\right]^2.$$

Both $\theta_1(\varphi)$ and $\theta_2(\varphi)$ are monotonically increasing functions of $\varphi$. We also have the bound

$$\theta(\varphi) \leq \frac{e^{\frac{\max \varphi}{u(0)+W}} - 1}{\frac{\max \varphi}{u(0)+W} - 1} = C_\theta.$$

Therefore, we get two more inequalities

$$\mathrm{Cov}(\theta_1(\varphi)\varphi, \varphi) \leq C_\theta \mathrm{Var}(\varphi), \quad \mathrm{Cov}(\theta_2(\varphi)\varphi^2, \varphi) \geq 0.$$

With all the preparatory works down, we give upper and lower bounds for $\mathrm{d}u/\mathrm{d}t$. We first upper-bound $\mathrm{d}u/\mathrm{d}t$.

$$\frac{\mathrm{d}u}{\mathrm{d}t} \leq \pi_v \mathrm{Cov}(e^{\frac{\varphi}{u+W}}, \varphi)$$

$$= \pi_v \mathrm{Cov}(1 + \frac{\theta_1(\varphi)\varphi}{u+W}, \varphi)$$

$$\leq \frac{\pi_v C_\theta \mathrm{Var}(\varphi)}{u}.$$

By solving the corresponding ODE, we get that

$$\frac{1}{2}u^2 \leq \sqrt{C_\theta \mathrm{Var}(\varphi)t} + C.$$

To give a lower bound, we have that

$$\frac{\mathrm{d}u}{\mathrm{d}t} \geq \frac{u(0)^2}{[u(0)+W]^2} \frac{\pi_v \mathrm{Cov}(e^{\frac{\varphi}{u+W}}, \varphi)}{\mathbb{E}e^{\frac{\varphi}{u(0)+W}}}$$

$$\geq \frac{u(0)^2}{[u(0)+W]^2} \frac{\pi_v}{\mathbb{E}e^{\frac{\varphi}{u(0)+W}}} \mathrm{Cov}(1 + \frac{\varphi}{u+W} + \theta_2(\varphi)\Big[\frac{\varphi}{u+W}\Big]^2, \varphi)$$

$$\geq \frac{u(0)^2}{[u(0)+W]^2} \frac{\pi_v}{\mathbb{E}e^{\frac{\varphi}{u(0)+W}}} \frac{\mathrm{Var}(\varphi)}{u+W}$$

$$\geq \frac{u(0)^2}{[u(0)+W]^2} \frac{\pi_v}{\mathbb{E}e^{\frac{\varphi}{u(0)+W}}} \cdot \frac{u(0)}{u(0)+W} \cdot \frac{\mathrm{Var}(\varphi)}{u}$$

$$= \tilde{C}_\theta \frac{1}{u}.$$

Therefore, $u \geq \sqrt{\tilde{C}_\theta t + \tilde{C}}$. In conclusion,

$$y_v = \log u = \frac{1}{2}\log t + r_v,$$

with $r_v$ bounded. $\qquad\square$

## B.4 VALUE STATE DRAINS IN THEOREM 3

**Theorem 10** (Restatement of Theorem 3). *Fixing $\boldsymbol{\alpha} = y\mathbf{1}$, $\boldsymbol{\beta} = c\mathbf{1} - e^{-\alpha}\boldsymbol{W} \circ \boldsymbol{\xi}$ with $c \in \mathbb{R}$. Define $\overline{\beta}(t) = V^{-1}\sum_{i=1}^{V} \beta_i(t)$. Then the gradient flow of $\boldsymbol{\beta}(t)$ converges:*

$$\boldsymbol{\beta}(t) \to \boldsymbol{\beta}^\star = \overline{\beta}(0)\mathbf{1} - e^{-\alpha}\boldsymbol{W} \circ \boldsymbol{\xi}.$$

*Proof.* Theorem 8 has already verified that $\boldsymbol{\beta} = c\mathbf{1} - e^{-\alpha}\boldsymbol{W} \circ \boldsymbol{\xi}$ are stationary points of loss. In the proof of Theorem 8, we have derived $\nabla_{\boldsymbol{\beta}}^2 \mathrm{loss}(\boldsymbol{\alpha}, \boldsymbol{\beta})$.

$$\nabla_{\boldsymbol{\beta}}^2 \mathrm{loss}(\boldsymbol{\alpha}, \boldsymbol{\beta}) = \sum_{k=1}^{V} \pi_k \mathbf{G}_k^{\mathbf{Q}}.$$

Lemma 5 indicates that it is positive semi-definite. Therefore, all stationary points attain the minimum of $\mathrm{loss}(\boldsymbol{\alpha}, \boldsymbol{\beta})$. Suppose $\boldsymbol{\beta}^\star$ is a stationary point, we therefore get that $q_{vk} = p_{vk}$ for any $v$, $k$. This implies that $e^y \beta_k^\star + W_k \xi_k$ are constants across $k$. We can solve $\boldsymbol{\beta}^\star$ and get that $\boldsymbol{\beta}^\star = c\mathbf{1} - e^{-\alpha}\boldsymbol{W} \circ \boldsymbol{\xi}$. The convexity of the $\mathrm{loss}(\boldsymbol{\alpha}, \boldsymbol{\beta})$ guarantees that $\boldsymbol{\beta}$ always converges to a stationary point $\boldsymbol{\beta}^\star$.

To find the value of $c$ in $\boldsymbol{\beta}^\star$, note that $\sum_{v=1}^{V} \dot{\beta}_v(t) = 0$. We get that $\overline{\beta}^\star = \overline{\beta}(0)$. Therefore, $\boldsymbol{\beta}^\star = \boldsymbol{\beta}^\star = \overline{\beta}(0)\mathbf{1} - e^{-\alpha}\boldsymbol{W} \circ \boldsymbol{\xi}$. $\qquad\square$

**Remark 11.** *If we assume that $p_{vk} > 0$ for any $v$, $k$ and suppose that the initial value $\boldsymbol{\beta}(0)$ is close enough to $\boldsymbol{\beta}^\star$, it is possible to prove the fast convergence of $\boldsymbol{\beta}(t)$ to $\boldsymbol{\beta}^\star$.*

$$\|\boldsymbol{\beta}(t) - \boldsymbol{\beta}^\star\|_2^2 \leq \delta e^{-\mu t}.$$

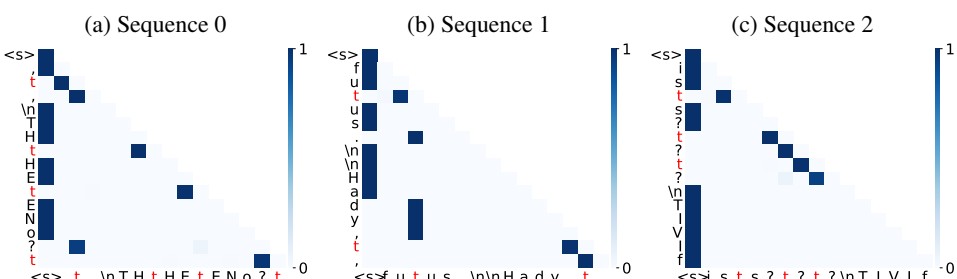

Figure 9: Attention plots of the one-layer transformer trained on the Bigram-Backcopy task.

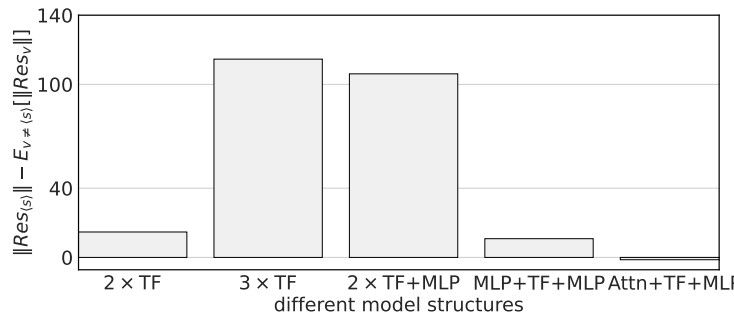

Figure 10: Minimal structures to elicit residual state peaks. We use $A + B + C$ to indicate the model with structure $A$, $B$, $C$ in layers 0, 1, and 2, respectively.

## C  ABLATIONS

**Experimental details.**    We train transformers with positional embedding, pre-layer norm, SoftMax activation in `attn`, and ReLU activation in `mlp`. We use Adam with constant learning rate 0.0003, $\beta_1 = 0.9$, $\beta_2 = 0.99$, $\varepsilon = 10^{-8}$, and a weight decay of 0.01. We choose a learning rate of 0.03 for the SGD. In each training step, we resample from the BB task with a batch size of $B = 512$ and sequence length $N = 256$. Unless otherwise specified, the model is trained for $10,000$ steps. Results are consistent across different random seeds.

**More attention plots**   : Figure 9 presents more attention-weight heat maps of the one-layer transformer model trained on the BB task. All attention maps show the attention sink phenomenon. Interestingly, the trigger tokens serve as attention sinks in some inputs.

### C.1  ABLATIONS OF DIFFERENT MODEL STRUCTURES TRAINED ON THE BIGRAM-BACKCOPY TASK.

**Exploring the minimal structure for massive norms.**    Figure 10 presents the difference of residual norms between the $\langle s \rangle$ token and others ($\|\text{Res}_{\langle s \rangle}\| - \mathbb{E}_{v \neq \langle s \rangle}[\|\text{Res}_v\|]$), with different combinations of model structures. The $3 \times \text{TF}$ and $2 \times \text{TF} + \text{mlp}$ are two outliers, showing clear evidence of residual state peaks.

**Attention plots, value state norms, and residual norms for a three-layer transformer trained on BB task.**    Figures 11, 12, and 13 show the extreme token phenomena in a three-layer transformer. The residual state peaks show different phenomena from those in LLMs, with the last layer output increasing the residual norms of non-$\langle s \rangle$ tokens. Figure 1 demonstrates that the residual state norms of $\langle s \rangle$ drop match the magnitudes of other tokens at the last layer.

**Statics and dynamics of the simplified model in Theorem 3.**    With the simplified model structure in Figure 4, we pre-train the model using Adam with learning rate 0.03. Figure 14 and 15 show results that match both the theory and the observations of the one-layer transformer.

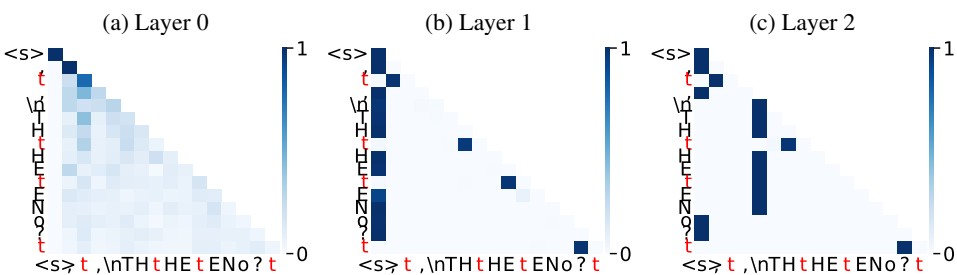

Figure 11: Value state norms of three-layer transformer trained on the BB task

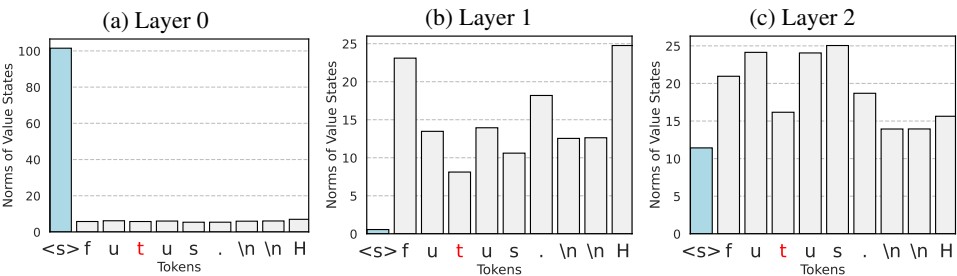

Figure 12: Value state norms of three-layer transformer trained on the BB task

## C.2 Variations of the Bigram-Backcopy task

**Bigram-Backcopy task without the ⟨s⟩ token.** We train a one-layer transformer on the BB task without the ⟨s⟩ token. Figure 16 shows that the ⟨s⟩ token is perhaps not the extreme token. Instead, trigger tokens and delimiter tokens seem to become extreme tokens. The results indicate that initial tokens may not be the only candidates for the extreme token, partially explaining why delimiter tokens could also be extreme tokens in LLMs.

**The Bigram-Skip-one (BS) task.** We make slight modifications to the Bigram-Backcopy task. On trigger tokens, instead of copying the preceding token, we sample from the bigram-probability of the preceding token $P(\cdot \mid \text{Second-to-last token})$. We train a one-layer transformer on it using the same configuration as the BB task. Figure 17 shows that extreme token phenomena are mitigated. The reason is that trained under BS, both the value states $\text{Val}_v$ and the token embedding $\text{ebd}_v$ give the logit of the bigram transition probability. Therefore, other than having attention sink on the ⟨s⟩ token, self-attention becomes a new possibility to achieve the active-dormant mechanism.

## D More Attention Heads in Dormant and Active Phase

In this section, we present two more dormant- and active- phase heads in Llama 2-7B-Base, in Figures 18 and 19, which are more difficult to interpret than Layer 16 Head 25, but go dormant on some inputs and active on others.

## E Fine-Grained Static Mechanisms for Extreme-Token Phenomena

In this section, we will identify more fine-grained static mechanisms for extreme-token phenomena in Llama 3.1-8B-Base. To do this, we identify circuits for the origin of attention sinks and small value states. Then, using ablation studies, we study the origin of massive norms. Again, we use the generic test phrase "⟨s⟩ Summer is warm. Winter is cold."

**Attention sinks and global contextual semantics.** There are many attention sinks at layer 0, and the ⟨s⟩ token is always the sink token (see Figure 20). From now on until the end of this section,

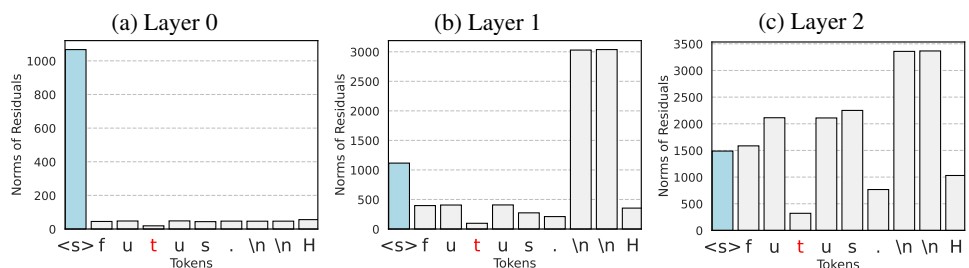

Figure 13: Residual state norms of three-layer transformer trained on the BB task

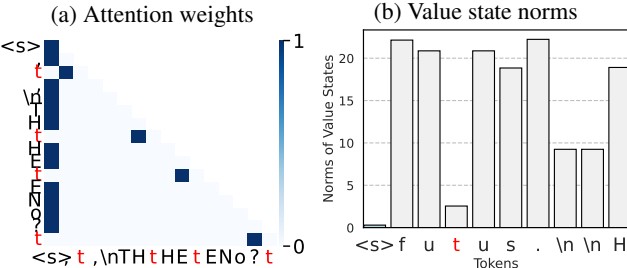

Figure 14: The simplified model structure trained on the BB task.

we *restrict our attention to Head 31 of Layer 0, which is an attention sink.* These attention sinks are caused by two linear-algebraic factors, demonstrated in Figure 21.

1. The key state of the ⟨s⟩ token has small dot product with all other tokens.

2. The query states of all tokens are nearly orthogonal to the key states of all tokens except the ⟨s⟩ token.

These two facts combine to ensure that the key state of the ⟨s⟩ token is picked out by each query state, causing the attention sink. Since these query and key states are produced without any cross-token interaction, the alignment of different states is caused purely by the token's global importance or meaning imparted via pretraining. The ⟨s⟩ token has no semantic meaning in the context of prose tokens, so its key state is not aligned with key states of meaningful prose tokens. Also, delimiter tokens, oft considered secondary attention sinks (c.f. Appendix F.2), have the most aligned key states to the key state of the ⟨s⟩ token, and are also the tokens with the least semantic meaning in the prose context. Thus, we identify that, at least in this restricted example, query state and key state alignment depends heavily on the contextual semantics of the token.

**Value state drains.** The value states of the ⟨s⟩ token at Layer 0 Head 31 are already near zero, as demonstrated in Figure 22. While the delimiter tokens, which are less semantically meaningful in the prose context, have smaller value states than the rest, they are not as small as the value state of the ⟨s⟩ token which is guaranteed to not have any semantics.

**Residual state peaks.** Residual state peaks are caused by the first two layers' MLPs. In particular, we perform several ablations, comparing between the residual state norms in a later layer (24) of an un-edited forward pass versus forward passes where we force the output of either multiple layers, a single layer, an attention block, or an MLP to be zero (and hence remove its contribution from the residual stream). This intervention showed that ablating *either* Layer 0's or Layer 1's MLP is sufficient to remove the residual state peak. In particular, the second-largest token at Layer 24 in *each* ablation (including the original setup) has norm between 29 and 38, so the interventions ensure that all tokens have similar size.

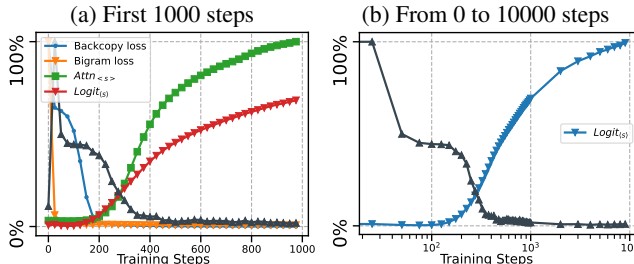

Figure 15: The dynamics of the simplified model structure trained on the BB task. *Left (a):* The training curves match the one-layer transformer. *Right (b):* The logit curve is close to the logarithmic growth predicted in Theorem 3.

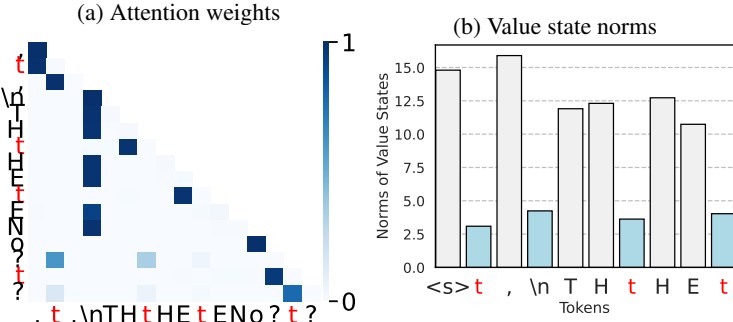

Figure 16: Attention weights and value state norms of a one-layer transformer trained on the BB task without the $\langle s \rangle$ token.

# F    ASSORTED CAVEATS

## F.1    MULTIPLE ATTENTION SINKS VS. ONE ATTENTION SINK

As we have seen, attention heads in the BB task (Section 2), Llama 2-7B-Base (Section 3.1), and OLMo (Section 3.2) exhibit multiple attention sinks. That is, when heads in these models are dormant, they tend to have two attention sinks. For the LLMs in this group, at least on prose data, the $\langle s \rangle$ token as well as the first delimiter token (e.g., representing . or ;) are sink tokens. Meanwhile, Llama-3.1-8B-Base (Section 3) only ever has one attention sink on prose data, and the $\langle s \rangle$ token is always the sink token. Here, we offer a possible explanation of this phenomenon. For the BB task, multiple sink tokens are necessary to solve the task. For LLMs, we believe this distinction may be explained by the relative proportion of coding data, in which delimiters have a greater semantic meaning than prose, within the training set. For instance, OLMo was trained on DOLMA (Soldaini et al., 2024), which has around 411B coding tokens. Meanwhile, Llama 2 used at most (2T ×0.08 =) 0.16T coding tokens. Finally, Llama 3.1 used around (15.6T × 0.17 =) 2.6T coding tokens (Dubey et al., 2024). On top of the raw count being larger, coding tokens are a larger proportion of the whole pre-training dataset for Llama 3.1 compared to other model families. Thus, during training, the presence of delimiters would not be considered unhelpful towards next-token prediction, since such delimiters carry plenty of semantics in a wide variety of cases. Our earlier hypothesis in Section 3.1 proposes that only tokens which lack semantics in almost all cases are made to be sink tokens. This could be a reason for the distinction.

## F.2    THE ROLE OF A FIXED $\langle s \rangle$ TOKEN IN THE ACTIVE-DORMANT MECHANISM

Some models, such as OLMo, are not trained with a $\langle s \rangle$ token. Despite this, the first token of the input still frequently develops into a sink token. We can study the effect of positional encoding of the tokens on the attention sink phenomenon by shuffling the tokens before inputting them into the transformer, and observing how and why attention sinks form. If we do this with the phrase "Summer is warm.  Winter is cold."  with OLMo, we observe that at Layer 24, there are many

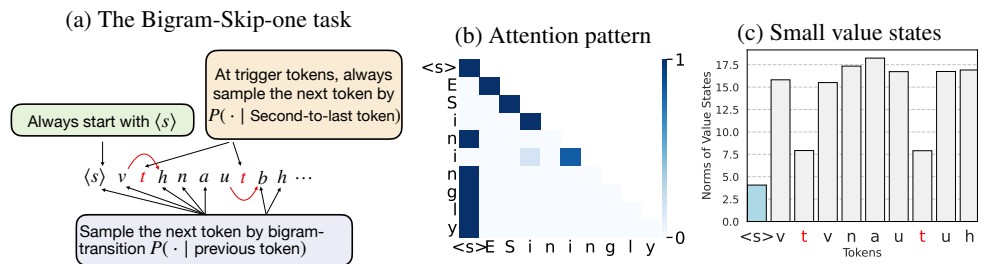

Figure 17: **Experiments on the Bigram-Skip-one task.** All phenomena are close to those in the BB task, but with diagonal attention sinks and relatively larger $\|\mathtt{Val}_{\langle s \rangle}\|$ compared with Figure 2.

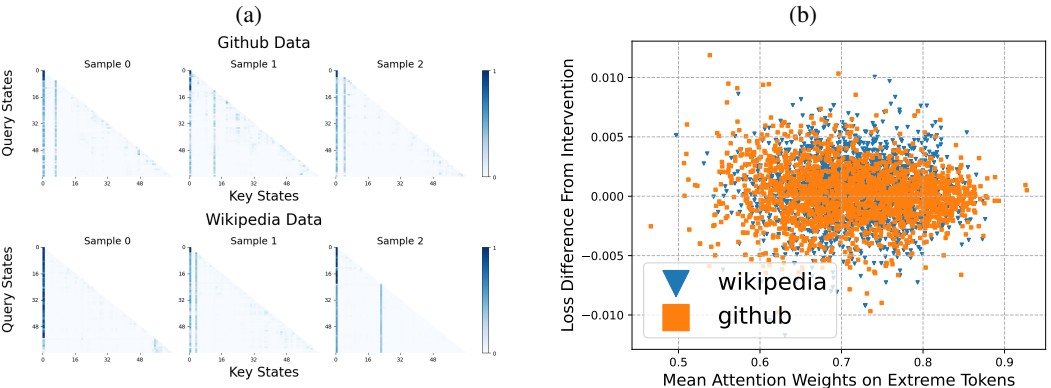

Figure 18: **Layer 16 Head 20 of Llama 2-7B-Base.**

attention sink heads where the first token and first delimiter token share attention mass, even if the sentence is jumbled up and makes no grammatical sense. This points towards the observation that without a $\langle s \rangle$ token, the attention sink formation uses both positional data and, to a greater degree, the semantic data of each token. We leave studying this effect in greater detail to future work.

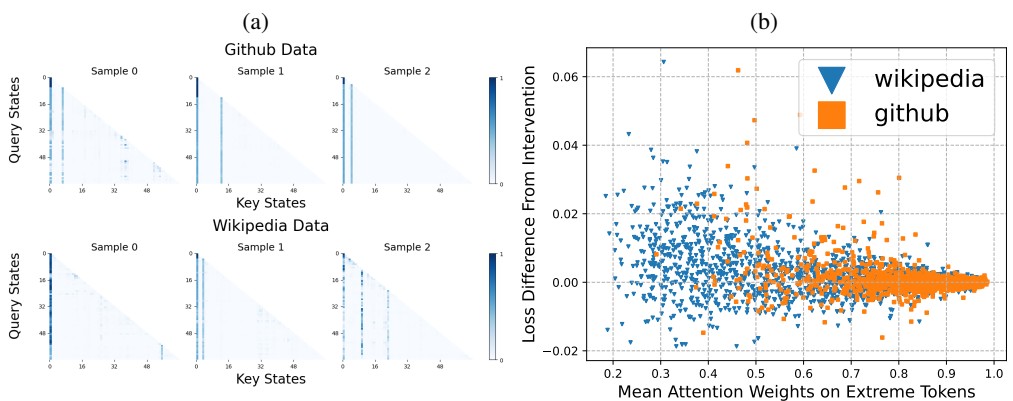

Figure 19: **Layer 16 Head 28 of Llama 2-7B-Base.**

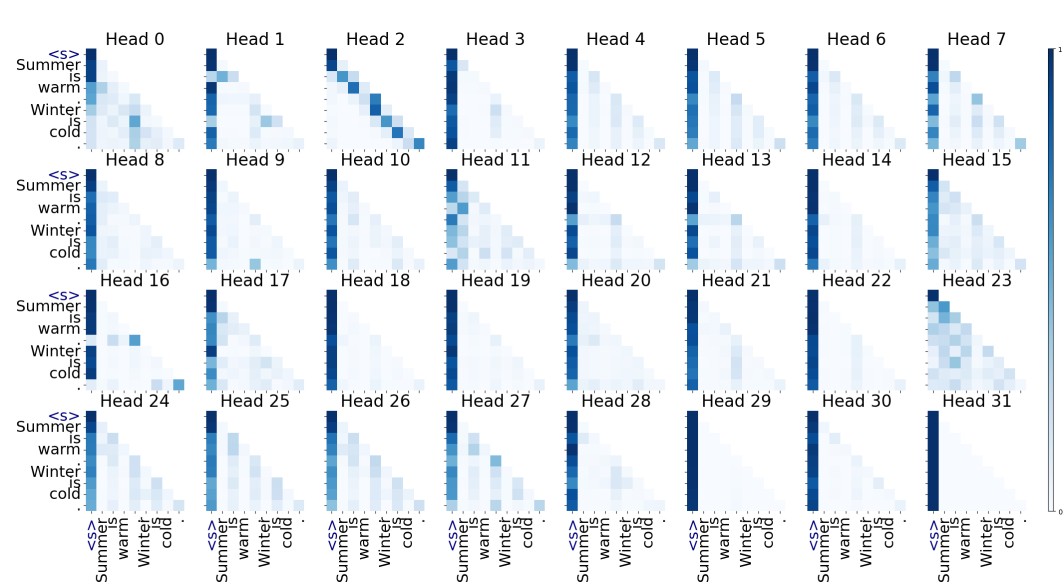

Figure 20: **A visualization of attention heads at Layer 0 of Llama 3.1-8B-Base.** Notice that many heads have the attention sink property, even at Layer 0 without any cross-token interaction. As usual, the test phrase is "Summer is warm. Winter is cold." The most clear attention sink is Head 31.

(a) Alignment of query states and key states (L0H31).    (b) Alignment of key states and key states (L0H31).

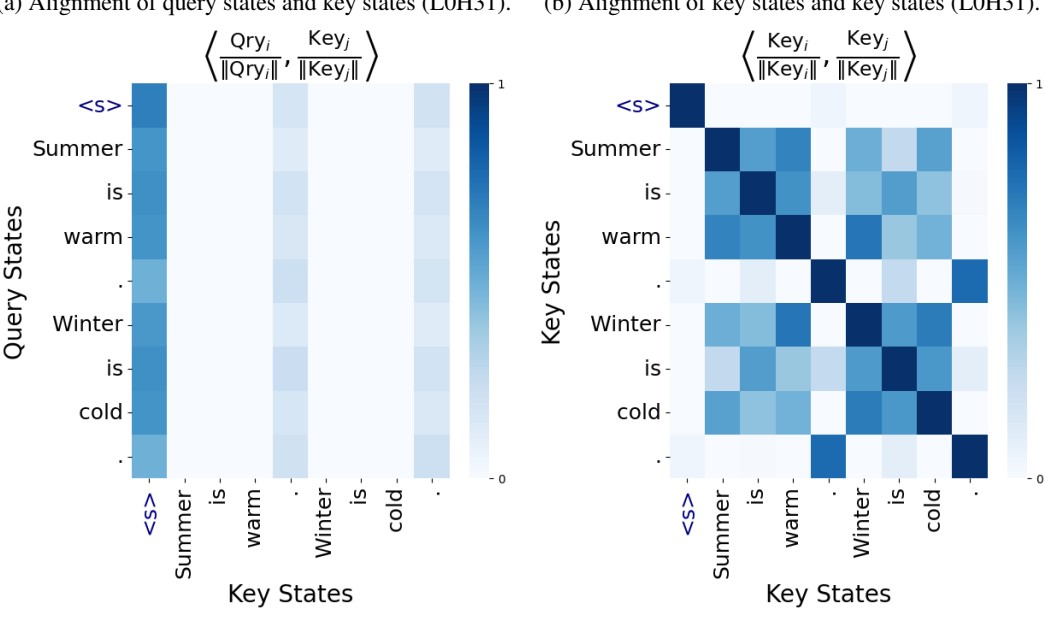

Figure 21: **Alignment between query states and key states at Layer 0 Head 31 of Llama 3.1-8B-Base.** We observe that the key state of $\langle s \rangle$ is orthogonal to all other key states, and heavily aligned with all query states. Meanwhile, all semantically meaningful (i.e., not delimiter) tokens have aligned key states.

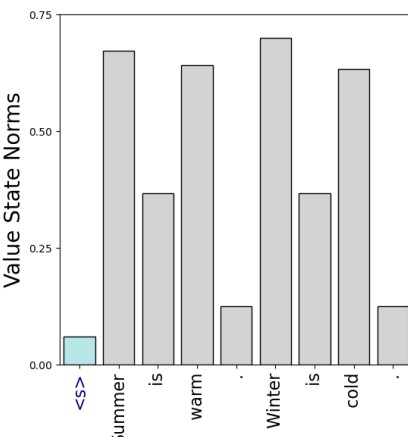

Figure 22: **Value state drains at Layer 0 Head 31 of Llama 3.1-8B-Base.** We observe that the value state associated with ⟨s⟩ is already much smaller than every other semantically meaningful token, and still smaller than the delimiter tokens in the same sentence.

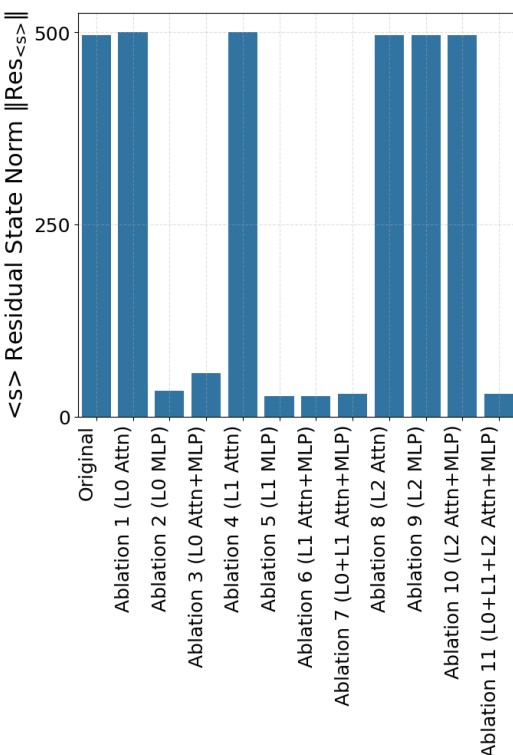

Figure 23: **Ablation study on the cause of the residual state peak in Llama 3.1-8B-Base.** We perform a series of ablations to understand which components of the network promote the residual state peaks. We find that ablating either the zeroth or first layer's MLP is sufficient to remove the residual state peak phenomenon, while no other layer-level ablation can do it.

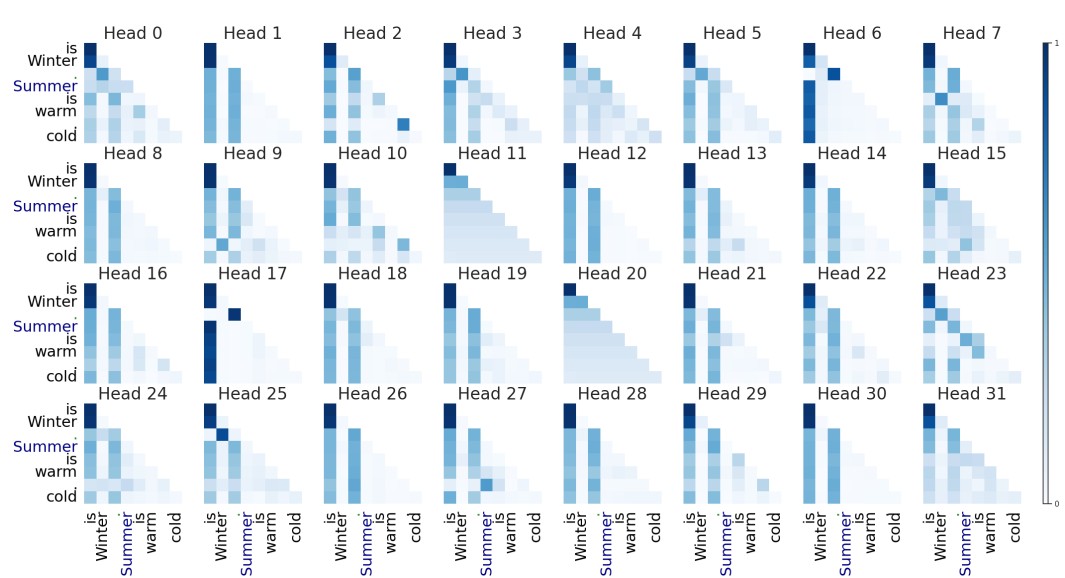

Figure 24: **Attention sinks with shuffled input in Layer 24 of OLMo.** In order to understand the impact of positional encodings when there is no ⟨s⟩ token, we shuffle the input of the test string "Summer is warm. Winter is cold." in OLMo. We observe that there is still an attention sink on token 0, despite it being a random token that does not usually start sentences or phrases (since it is uncapitalized). This shows that the positional embedding, say via RoPE, has a large impact on the formation of attention sinks — when the semantics of each token have switched positions, the attention sink still forms on the zeroth token.

