# OpenReview forum: "Active-Dormant Attention Heads: Mechanistically Demystifying Extreme-Token Phenomena in LLMs"
_ICLR.cc/2025/Conference — Submitted to ICLR 2025_

### Official Review · Reviewer_CmR2 · 2024-10-29

**Soundness:** 3
**Presentation:** 3
**Contribution:** 3
**Rating:** 6
**Confidence:** 3

**Summary:**

This paper examines the mechanisms behind the "extreme-token phenomena" in large language models (LLMs), such as attention sinks, value-state drains, and residual-state peaks. These phenomena are collectively attributed to the "active-dormant" mechanism in transformer-based LLMs. The study first demonstrates these phenomena on simpler models trained on the Bigram-Backcopy (BB) task, identifying a mutual reinforcement dynamic driving these behaviors. The authors then extend their analysis to pre-trained LLMs (e.g., Llama and OLMo) and suggest mitigation techniques, such as replacing SoftMax with ReLU or switching optimization methods to control these extreme-token phenomena.

**Strengths:**

1. The study makes clear contributions by identifying the active-dormant mechanism and mutual reinforcement dynamic, helping to explain complex behaviors in transformer-based models.

2. The paper rigorously combines theoretical characterizations with empirical validation, making its claims robust and well-supported.

**Weaknesses:**

1. There remains a gap between the BB Task and real language scenarios. Although I recognize the potential practical value of the authors' intuitive explanation of these phenomena, I would still expect authors to validate their explanations in real scenarios. For example, it would be helpful to confirm whether deactivating attention heads with attention sinks truly has no impact on the final output.

2. The paper’s explanation for why Adam tends to cause residual state peaks is insufficient, as it mainly describes the phenomenon without deeper analysis. Furthermore, I did not see an evaluation of the impact of applying SGD and ReLU to real language models. In my opinion, both SGD and ReLU come with certain drawbacks, which should be discussed.

3. Here is a potentially relevant study [1] where the "broadcasting" mechanism mentioned has a formation process similar to the attention sink.

[1] Anchor function: a type of benchmark functions for studying language models. arXiv:2401.08309.

**Questions:**

See weaknesses.

---

> ### Author Response · Authors · 2024-11-22
>
> We thank the reviewer for their insightful comments. We respond to the questions as follows:
>
> >For example, it would be helpful to confirm whether deactivating attention heads with attention sinks truly has no impact on the final output.
>
> We agree with this point. Figure 6(b) shows the results of deactivating (zeroing out) attention heads with attention sinks. In the Wikipedia domain, the head exhibits a pure attention sink, and deactivating it has minimal impact on the final output.
>
> >The paper’s explanation for why Adam tends to cause residual state peaks is insufficient, as it mainly describes the phenomenon without deeper analysis.
>
> The intuition is that the residual state of the extreme token tends to have a smaller gradient as the training dynamics converge. However, Adam makes small gradients to become constant updates, leading to a linear increase in the norm of the residual states. We will provide more details with simulations in the revised version.
>
> >Here is a potentially relevant study [1] where the "broadcasting" mechanism mentioned has a formation process similar to the attention sink.
>
> Thanks for sharing. The active-dormant mechanism is ubiquitous in transformers. We will make more discussions of the related mechanism observed in other works in the revised version.

---

> > ### Comment · Reviewer_CmR2 · 2024-11-26
> >
> > Thank you for your detailed response and the additional experiments. I have no further questions and will maintain my score in favor of acceptance.

---

### Official Review · Reviewer_F3q2 · 2024-11-02

**Soundness:** 2
**Presentation:** 3
**Contribution:** 2
**Rating:** 5
**Confidence:** 3

**Summary:**

This paper examines the "attention sink" phenomenon in large language models, focusing on three related concepts: attention sinks, value-state drains, and residual-state peaks. Through a toy Bigram-Backcopy task, the paper reveal an active-dormant mechanism in which the attention head assigns dominant weight to the initial token in the dormant phase and shifts focus to relevant context tokens in the active phase. They attribute this behavior to a mutual reinforcement mechanism. Extending this analysis to real large language models, the authors show that this mechanism also emerges during pre-training, suggesting a understanding of the issue and potential mitigation strategies.

**Strengths:**

- The paper addresses the attention sink phenomenon in large language models, an important and timely issue.
- The analysis using a toy example is original, providing intuitive insights and interpretations of the problem.
- The motivation is clear and well-articulated.
- The paper is well-organized, and the figures are highly illustrative.

**Weaknesses:**

My primary concern is the statistical significance of the observed results, which raises questions about the validity of the conjecture regarding the active-dormant mechanism. For instance, in the BB task toy experiments, a simplistic generation rule (bigram transitions) is assumed, making the behavior of trigger and non-trigger tokens very distinct. This makes it difficult to accept that the theory and conjecture are applicable to real large language models, where such clear distinctions between tokens (trigger vs. non-trigger) do not exist. Section 3, a specific attention head in a specific layer is shown to exhibit behavior similar to the conjectured mechanism from Section 2. However, I find this unconvincing, as demonstrating a single head is insufficient to establish that the active and dormant phases align with what was observed in Section 2 for the same reasons.

My second concern is the lack of demonstrated practical benefits from this analysis. Although the authors suggest that this study could offer potential mitigation strategies, the paper does not validate the effectiveness of these mitigations, which limits the overall impact and applicability of the work.

**Questions:**

- What is the motivation for using the BB task and designing the trigger and non-trigger tokens?
- On Lines 80 and 87, the word "mechanism" appears twice consecutively, which seems to be a typo.
- In Claim 4, could you clarify the case where "adding any value state from previous tokens worsens the prediction"?
- Why does the causal mask cause training dynamics to favor the <s> token? In Figure 6, attention is also given to token 3, which is described as semantically unmeaningful. So, is the cause of the attention sink truly the causal mask, or is it the lack of semantic meaning in certain tokens?

---

> ### Author Response · Authors · 2024-11-22
>
> We thank the reviewer for their insightful comments. We respond to the questions as follows:
>
> >Section 3, a specific attention head in a specific layer is shown to exhibit behavior similar to the conjectured mechanism from Section 2. However, I find this unconvincing, as demonstrating a single head is insufficient to establish that the active and dormant phases align with what was observed in Section 2 for the same reasons.
>
> We acknowledge the limitation of demonstrating the attention pattern for only one head. However, brute-force searching for attention patterns across active and dormant phases for all heads is computationally impractical.
> Instead, the training dynamics of OLMo provide alternative evidence for the active-dormant mechanism. Since the dynamics of attention sinks and value state drains consistently follow the mutual reinforcement mechanism predicted by the theory, it is likely that active-dormant mechanism is the driving force of the extreme-token phenomena in LLMs. We believe that dynamics analysis offers a stronger and more efficient method for demonstrating the active-dormant mechanism across all attention heads.
>
> >My second concern is the lack of demonstrated practical benefits from this analysis.
>
> We want to emphasize the practical significance of understanding extreme-token phenomena. Numerous studies have shown that these phenomena can cause issues in quantization, interpretability, and long-context inference [Xiao et al., 2023; Bondarenko et al., 2023], leading to the development of mitigation strategies.
> Our analysis provides insights into two key practical questions:
> Are extreme-token phenomena beneficial or harmful to the statistical performance of LLM?
> We found that these phenomena are neither inherently beneficial nor harmful; they represent a dormant phase of attention heads, and their elimination does not affect the statistical performance of transformers.
> What causes the extreme-token phenomena? By analyzing the model structure and optimizer, we identify SoftMax and ReLU as two key reasons for the extreme-token phenomena.
> We agree that validating the effectiveness of mitigation strategies is crucial and represents a promising direction for future research. While we do not explore these strategies fully in this work, we believe our work lays a solid foundation for further investigation into this phenomenon.
>
> >What is the motivation for using the BB task and designing the trigger and non-trigger tokens?
>
> Finding the minimal condition to recapitulate extreme-token phenomena is crucial for further understanding of them. The BB model can produce all extreme-token phenomena and bear close resemblance with LLMs, providing a nice framework for further studying the phenomena.
> The trigger token corresponds to the “non-sink” phase of attention heads, and the “non-trigger” tokens correspond to the “sink” phase. The BB task is designed so that a pre-trained attention needs to switch between “non-sink (active)” and “sink (dormant)” given different inputs.
>
> >In Claim 4, could you clarify the case where "adding any value state from previous tokens worsens the prediction"?
>
> In the BB model, the transformer needs to memorize the Bigram transition probabilities of non-trigger tokens, which only require the information from the current token. Including previous tokens in this process introduces noise, making memorization more challenging. Consequently, the attention mechanism remains dormant on non-trigger tokens.
>
> >Why does the causal mask cause training dynamics to favor the <s> token?
>
> Take the token on the second position as an example. Due to the causal mask, the <s> is the only token that it can sink its attention weights to. For the third, fourth, and following tokens, they all share <s> as the sink token. Therefore, <s> tends to become the extreme token.
>
> >So, is the cause of the attention sink truly the causal mask, or is it the lack of semantic meaning in certain tokens?
>
> Our intuition is that both of them contribute to the attention sink. In real LLMs, both semantic meaningless tokens (delimiters) and <s> would be attention sinks. We also observe similar phenomena in the BB model. Results are in Appendix F.
>
> References
>
> Xiao, Guangxuan, Yuandong Tian, Beidi Chen, Song Han, and Mike Lewis. "Efficient streaming language models with attention sinks." arXiv preprint arXiv:2309.17453 (2023).
>
> Bondarenko, Yelysei, Markus Nagel, and Tijmen Blankevoort. "Quantizable transformers: Removing outliers by helping attention heads do nothing." Advances in Neural Information Processing Systems 36 (2023): 75067-75096.

---

> > ### Comment · Reviewer_F3q2 · 2024-11-26
> > **Thank you for your response**
> >
> > Thank you for your response. However, my concerns are still not fully addressed. Could you please elaborate further on the following points? I feel the conjecture is somewhat elegant but has not been thoroughly examined:
> > > For instance, in the BB task toy experiments, a simplistic generation rule (bigram transitions) is assumed, making the behavior of trigger and non-trigger tokens very distinct. This makes it difficult to accept that the theory and conjecture are applicable to real large language models, where such clear distinctions between tokens (trigger vs. non-trigger) do not exist.
> >
> > While I understand that brute-force searching for attention patterns across active and dormant phases for all heads is computationally impractical, analyzing only a single head is insufficient to draw robust conclusions. A larger fraction of heads should at least be considered to support the claims.
> >
> > Regarding the practical benefits, thank you for the elaboration. Could you provide a simple justification or example of what potential mitigation strategies could arise from this paper?

---

> > > ### Author Response · Authors · 2024-12-02
> > >
> > > We thank the reviewer for their valuable feedback. Below, we provide more detailed explanations addressing the comments.
> > >
> > > > How does the trigger token assumption in the BB model apply to LLMs?
> > >
> > > In LLMs, trigger tokens depend on the context. For example, the induction head, which has been consistently observed across LLMs, activates on repeated tokens and predicts the subsequent token based on the token that followed the earlier occurrence.
> > >
> > > For instance, given the input “ABA,” the second occurrence of “A” serves as the trigger token for the induction head. At this point, the induction head assigns attention weights to the preceding “B” and predicts “B” as the next token.
> > >
> > > Conversely, given the input “CBA,” the first “A” is removed from the sequence, making the second “A” a non-trigger token. As a result, the induction head remains dormant.
> > >
> > > Under different contexts, the second "A" may or may not become a trigger. Since we only focus on analyzing one head, we fix a small group of tokens as triggers and ignore the triggers' dependence on the context.
> > >
> > >  > Showing the active-dormant mechanism for more attention heads.
> > >
> > > We agree that demonstrating the mechanism for a single head is insufficient. To address this, we have extended our analysis in two ways:
> > >
> > > 1. Analysis of Multiple Heads and Their Dynamics: We show that multiple heads in OLMo exhibit similar dynamics to the BB model (Figures 7 and 8). As discussed in Section 2, the BB model’s dynamics are governed by the active-dormant mechanism. Since the dynamics of LLMs align with those of the BB model, we infer that LLMs share the same active-dormant mechanism. This provides an evidence supporting our hypothesis.
> > > 2. Zeroing-Out Intervention Across Attention Heads: We apply a zeroing-out intervention under the Wiki/Github domains for all heads in Layer 16 of Llama-2-7B. The results, presented in Figure 19, reveal trends consistent with the behavior of the specific head analyzed in the main paper. We dropped them because they do not show clear separation as the special head shown in the main paper.
> > >
> > >  > Could you provide a simple justification or example of what potential mitigation strategies could arise from this paper?
> > >
> > > One possible mitigation strategy involves introducing special register tokens. This approach preserves the active-dormant mechanism while mitigating the effects of extreme tokens in the original model. Sun et al. [2023] successfully employed this strategy to address similar issues.
> > >
> > > Sun, Mingjie, Xinlei Chen, J. Zico Kolter, and Zhuang Liu. "Massive activations in large language models." arXiv preprint arXiv:2402.17762 (2024).

---

### Official Review · Reviewer_NsbM · 2024-11-02

**Soundness:** 2
**Presentation:** 3
**Contribution:** 3
**Rating:** 6
**Confidence:** 3

**Summary:**

The paper studies the emergence of extreme-token phenomena in modern transformer neural networks, namely: Attention sinks, Value state drains and Residual state peaks, all previously identified in recent works. The first refers to the observation that in some attention heads, special tokens named sink tokens attract a large proportion of the attention weights. The second highlights that values states associated with sink tokens are usually significantly smaller than those of other tokens. The third refers to the observation that the intermediate representations (except for initial and final layers) of sink tokens are characterised by larger norms compared to those of other tokens.


The authors first show that attention sinks and value state drains can be identified in a simple settings: one layer transformer models trained on the Bigram-Backcopy task which involves sequences generated either through bigram transitions or by copying previous tokens in the presence of special trigger tokens. Through an in-depth empirical analysis, the authors show that all non-trigger tokens exhibit attention sinks, while the attention for trigger tokens is concentrated on their preceding positions. In particular, the BOS tokens attracts a significant proportion of the attention weights for the non-trigger tokens, and its value state decreases in norm accordingly. The authors highlight that the aforementioned extreme token phenomena develop reinforcing each other during training. This observation is corroborated by a theoretical analysis showing that indeed "Attention logits grow logarithmically reinforced by small value states" and "Value state shrinks to a small constant vector reinforced by large attention logits".

The final part of the paper is devoted to an investigation on the emergence of these phenomena in LLMs, with a particular focus on Llama 2-7B-Base and  OLMo-7B, both open-source. On the first model, a specific head is described in terms of its behaviour when the model is prompted with two different data sources. A very similar mechanism is shown to emerge, with the head being dormant on Wikipedia data (prose) as opposed to Github (code) data. Thanks to the availability of intermediate checkpoints for OLMo-7B, the authors are able to draw a parallel between their theoretical analysis of the learning dynamics of their toy model and that of OLMO, resulting in a good agreement.

**Strengths:**

- The paper is very well written and easy to follow.

- Extreme-Token Phenomena are very interesting and quite common across LLMs. The paper provides a simplified setting to theoretically describe their emergence. In particular, the authors show that these phenomena manifest themselves in synergy through training, via "mutual reinforcement dynamics".

- The paper shows that the analysis developed on the BB task extends to some extent to LLMs. In particular, by manually deleting a specific attention head on the LLAMA model, the authors show that the resulting model's performance depends on whether the head is dormant or not, which in turn depends on the specific input context (Fig. 6). In addition, Fig. 7 and 8 corroborate the theoretical results on the simple BB model.

**Weaknesses:**

- In section 2.1 (see Figure. 4), the considered one-layer transformer does not alternate the attention layer and the MLP along the depth dimension. Rather it uses these two components in parallel. The reason for this choice is not clear and should be better motivated. In addition, the trained models use pre-layer norm and it is not clear if residual connections are used and how.

- In the derivation of the loss(alpha, beta), it is not clear to me how the stable distribution is defined. Could you clarify?

- **Replacing SoftMax by ReLU attention removes extreme-token phenomena**: Can you provide more intuition on the role of the softmax? Its utilisation appears to be correlated with the emergence of extreme-token phenomena. The natural question then is whether this component is actually needed in practice and why modern LLMs use it. Could you elaborate on this?

- A question on Theorem 3, first point: I would have expected the logits to grow logarithmically as beta decreases (i.e. small value states). Instead, no constraint on beta seems to be needed for the logits to grow. I would appreciate it if the authors could clarify this point.

- As for the emergence of residual peaks, the experiments show that 3 layers are needed for the residual peaks to appear. First, are the authors now using a standard transformer architecture with attention layers interleaved with mlps? Second, do the authors have any intuition on why at least 3 layers are needed?

- In the construction of the token embeddings, the authors are assuming them to have dimension V, i.e. they are represented as one-hot vectors. This implies different tokens are orthogonal to one another. How important is this assumption?

-  In the BB example, it is shown that a head can have sinks yet being non-dormant (due to trigger tokens). Is this case contemplated by the theory? This seems to be quite general and is reflected in real LLMs (fig. 18 and fig. 19 in the appendix). To this extent, fig. 6 seems to be a pretty special case, where, for wikipedia data, the head is actually dormant as there are only two sink tokens and nothing else.

- How did you practically identify the heads exhibiting sink tokens in real LLMs?

- Are the results obtained for OLMO also valid for Pythia (for which intermediate checkpoints are also available)?

**Questions:**

See weaknesses part.

---

> ### Author Response · Authors · 2024-11-22
>
> We thank the reviewer for their insightful comments. We respond to the questions as follows:
>
> >Rather it uses these two components in parallel. The reason for this choice is not clear and should be better motivated.
>
> The motivation for the parallel structure comes from the interventions in the 1-layer BB model (Figure 3(a)), which show that the attention and MLP layers have separable utilities. Even when trained with a sequential structure, the attention layer predicts the next token only for trigger tokens, having no impact on non-trigger tokens. Conversely, the MLP layer predicts the next token exclusively for non-trigger tokens. As a result, the outputs of the attention and MLP layers belong to different subspaces. This indicates that the trained sequential structure is functionally equivalent to a parallel structure.
>
> >In addition, the trained models use pre-layer norm and it is not clear if residual connections are used and how.
>
> We use residual connections and sequential structure for the trained models. We will add more details in the revised version.
>
> >In the derivation of the loss(alpha, beta), it is not clear to me how the stable distribution is defined. Could you clarify?
>
> The stable distribution is defined as the marginal distribution of tokens in the BB task, which means the frequency of each token in the sequence generated by the BB task.
>
> >Can you provide more intuition on the role of the softmax?
>
> The SoftMax requires the total attention weights from each token to previous tokens to sum up to 1. However, sometimes the previous token information is unnecessary for the next word prediction. In this situation, more attention weights will concentrate on tokens with small value states, leading to the extreme-token phenomena.
>
> >The natural question then is whether this component is actually needed in practice and why modern LLMs use it. Could you elaborate on this?
>
> We agree that understanding whether softmax is essential is an important question, one that warrants a comprehensive analysis in future work. -. Some work has shown that a SoftMax transformer may be stabler in the training and has better performance [Shen et al., 2023; Hua et al., 2022].
>
> >First, are the authors now using a standard transformer architecture with attention layers interleaved with mlps? Second, do the authors have any intuition on why at least 3 layers are needed?
>
> For the first question: Yes. For the second: Our intuition is that the “OV” structure in the attention plus the MLP of the first layer is essential to the rise of massive residual states. The third layer MLP, however, removes the massiveness of the residual states so that it doesn’t hurt the read-out layer. We do not include them as we lack appropriate experiments to support the intuition. A more fine-grained analysis is an interesting future direction.
>
> >In the construction of the token embeddings, the authors are assuming them to have dimension V
>
> Since the output of the transformer should have dimension $V$, we set the hidden dimension $d=V$ to avoid complications. All results can generalize to arbitrary embedding dimension $d$ if we include the $V*d$ read-out layer in the theoretical model.
>
> > it is shown that a head can have sinks yet being non-dormant (due to trigger tokens). Is this case contemplated by the theory?
>
> Yes. Our theory only considers non-trigger tokens and focuses on explaining the sinks. The trigger tokens would have all the attention weights concentrated on the previous token.
>
> >How did you practically identify the heads exhibiting sink tokens in real LLMs?
>
> One way to identify the heads which exhibit sink tokens for a particular sample is to first identify which tokens have the potential to become sink/extreme tokens in a particular head. We can do this by examining the residual state peak phenomenon: the residual state norms of the (potential) sink tokens are significantly greater (usually > 10x greater) than others. Then, we examine heads which have large mean attention score on the sink tokens, i.e., compute the average attention score $\frac{1}{n-i}\sum_{j = i}^{n}A_{hij}$ for each token $i$ at head $h$ and examine the attention score on the potential sink tokens identified in the previous step. If these scores are large then the head has sink tokens, otherwise it does not. This is a straightforward and intuitive method to find such heads, but we do not claim it is optimal in any way; we leave improving this mechanism to future work.
>
> References
>
> Shen, Kai, Junliang Guo, Xu Tan, Siliang Tang, Rui Wang, and Jiang Bian. "A study on relu and softmax in transformer." arXiv preprint arXiv:2302.06461 (2023).
>
> Hua, Weizhe, Zihang Dai, Hanxiao Liu, and Quoc Le. "Transformer quality in linear time." In International conference on machine learning, pp. 9099-9117. PMLR, 2022.

---

### Official Review · Reviewer_JE77 · 2024-11-03

**Soundness:** 3
**Presentation:** 3
**Contribution:** 2
**Rating:** 6
**Confidence:** 3

**Summary:**

The paper investigates the underlying mechanisms of "extreme-token phenomena" in large language models (LLMs), characterized by three main effects: attention sinks, value-state drains, and residual-state peaks. These effects result in specific tokens persistently attracting high attention weights (attention sinks), displaying reduced value states (value-state drains), and exhibiting notably large norms in intermediate representations (residual-state peaks). The authors explore this by analyzing transformer models in a controlled setting, the Bigram-Backcopy (BB) task, identifying an active-dormant mechanism where attention heads behave differently based on input tokens. They find that mutual reinforcement between attention sinks and value-state drains drives these phenomena, with residual state peaks emerging in deeper architectures. They also find that attention sinks and value state draining are related, and interaction between the mutual reinforcement mechanism and Adam is responsible for residual state peaks. Expanding from a toy model's theoretical and empirical analysis, the paper shows that an active-dormant mechanism controls attention heads. Specifically, it highlights the active-dormant mechanism in existing pre-trained LLMs like Llama2 and OLMo-7B-0704 by identifying an interpretable active-dormant head through causal intervention analyses. By intervening in attention mechanisms and optimization strategies, such as replacing SoftMax with ReLU activations, they demonstrate methods to mitigate these extreme-token phenomena.

**Strengths:**

-	The paper provides valuable insights into LLMs' underlying mechanisms of extreme-token phenomena (attention sinks, value-state drains, and residual-state peaks). By identifying the active-dormant mechanism in attention heads and a mutual reinforcement process, the paper contributes to a deeper understanding of transformer model behavior, enhancing interpretability in LLMs.
-	The paper provides a comprehensive theoretical and empirical analysis of the mutual reinforcement mechanism in a toy one-layer transformer model to explain extreme-token phenomena. The identification of mutual reinforcement dynamics is robustly supported by theoretical analysis and validated through experiments on pre-trained LLMs using the Bigram-Backcopy task, offering strong evidence that this mechanism underlies attention sinks, value-state drains, and residual state peaks in both simplified and complex models.
-	The paper’s application of findings from toy models to analyze real-world LLMs, such as Llama2 and OLMo, significantly strengthens the study's impact. Validating the mutual reinforcement mechanism and active-dormant attention head dynamics in large-scale models highlights these identified mechanisms' broader applicability and relevance. This cross-model validation not only supports the robustness of the findings but also enhances the study's generalizability and potential utility for advancing transformer-based architectures

**Weaknesses:**

-	The motivation behind the mitigation of extreme-token phenomena is not clear. It is not clearly established that extreme token adversely affects the model’s performance. Although the paper explores interventions like replacing SoftMax with ReLU, these modifications are limited and could unintentionally impact model performance. The study doesn’t extensively examine trade-offs associated with these changes, such as how they might affect other aspects of model performance or generalization ability. This leaves questions about whether mitigating these phenomena would lead to meaningful improvements in model usability or performance in practical applications. Could the authors provide a more comprehensive explanation or analysis of why or what specific effects reduction of the phenomena on the downstream task?
-	The paper claims that the formation of extreme tokens in LLMs follows a similar mutual reinforcement mechanism as observed in a simplified BB model, with theoretical results that can be generalized to more complex models. However, this justification relies on specific assumptions, including zeroing out attention heads for intervention outcomes, which will not impact token prediction accuracy. This assumption may oversimplify the interactions in real-world LLMs, where dependencies between heads and layers are often more intricate. Additionally, while some empirical evidence in pre-trained models aligns with the theory (active and dormant attention heads within Llama2), the scope of models evaluated is limited, raising questions about the robustness of the claim across diverse architectures and tasks. Would it be possible for authors to show whether their claims about specific attention heads being active and dormant but under the premise that context are not significantly different? For example, mathematics-based context vs language comprehension-based context. It would help gauge the sensitivity of the claims when the contexts are not vastly different. In particular, would we still find attention heads with simple and interpretable active behavior for weaker contextual differences?

**Questions:**

-	In section 2.3, it is mentioned that no residual peaks appear in a one-layer transformer(BB model) or even with different combinations (2xTF, MLP+TF+ML, Attn+TF+MLP); it would be interesting to know why it seems to have residual peaks in certain instances. The authors claim that with a BB model, “residual state peaks contribute to the attention sink and value state drain phenomena.” However, it still remains unclear between the dependency between residual state peaks and, attention sink and value state drain. It would be helpful to see the comparison of residual state peaks and attention sink and value state drain phenomena for the 1-layer BB model even when residual state peaks do not appear.

---

> ### Author Response · Authors · 2024-11-22
>
> We thank the reviewer for their insightful comments. We respond to the questions as follows:
>
> > The motivation behind the mitigation of extreme-token phenomena is not clear.
>
> Thank you for pointing out the need to clarify the motivation for studying extreme-token phenomena and their mitigation. The primary goal of our paper is not to propose or advocate for mitigation strategies but rather to provide a theoretical understanding of why extreme-token phenomena arise in the first place.
>
> While our paper explores potential interventions like replacing SoftMax with ReLU, these modifications were used primarily as tools to examine the mechanisms behind the phenomena, rather than as definitive solutions to improve downstream performance. Consequently, we have not exhaustively explored the trade-offs associated with these changes. We agree that mitigating extreme-token phenomena would be an important area for future research.
>
> >This assumption may oversimplify the interactions in real-world LLMs, where dependencies between heads and layers are often more intricate.
>
> We agree that our theory simplifies many of the complex interactions present in real-world LLMs. However, we consider this simplification a strength rather than a limitation. By distilling the mechanism to its core, our theory isolates and highlights the essence of the mutual reinforcement mechanism that underlies the formation of extreme tokens. Developing a theory within an over-complicated system would risk hiding the key insights and limiting its general applicability.
>
> What is particularly compelling about our framework is that the behavior of the simplified model, which demonstrates the mutual reinforcement mechanism, aligns closely with the training dynamics observed in OLMo, a real-world LLM with a more intricate structure. This connection suggests that the simplified theory successfully captures the foundational dynamics relevant to more complex systems.
> Additionally, the assumption that “zeroing out attention heads for intervention outcomes will not impact token prediction accuracy” has been empirically validated in several attention heads of Llama-2, further supporting the practical relevance of our theoretical framework.
>
> >In particular, would we still find attention heads with simple and interpretable active behavior for weaker contextual differences?
>
> We agree that it would be interesting to discover simple and interpretable active-vs-dormant behavior for more subtle contextual differences. The example we have in the paper is to illustrate the point that the theoretical principles carry over to large language models. In particular, we do not guarantee that every domain difference corresponds to a head with similar behavior to the head in the paper. As such, we leave the discovery of such heads for particular interesting contexts to future work.
>
> >In section 2.3, it is mentioned that no residual peaks appear in a one-layer transformer(BB model) or even with different combinations (2xTF, MLP+TF+ML, Attn+TF+MLP); it would be interesting to know why it seems to have residual peaks in certain instances.
>
> We have three intuitions:
>
> 1. The First Layer Creates the Massive Residual State: The first attention and MLP layer have large outputs on the $\langle s \rangle$ token, while keeping other residual states to be small.
>
> 2. The Middle Layers Have Minimal Effect on Massive Residual States: The intermediate layers appear to contribute little to the magnitude of the residual states.
>
> 3. The Final Layer MLP Mitigates the Residual State Peak: The third-layer MLP reduces the residual state of the $\langle s\rangle$ token. Without this adjustment, an excessively large on the $\langle s\rangle$ token causes troubles in the read-out layer.
>
> Although we have not included these intuitions in the main analysis due to a lack of rigorous experimental support, we consider a more fine-grained investigation of these ideas an important future direction.
>
> > It would be helpful to see the comparison of residual state peaks and attention sink and value state drain phenomena for the 1-layer BB model even when residual state peaks do not appear
>
> The residual state peaks strengthen the attention sink and the value state drain in upper layers. The intervention results (Figure 5(b)) show that the residual state peaks lead to attention sink and value state drain in upper layers. Further comparing the extreme-token phenomena in one-layer and three-layer transformers, we have that:
>
> |          | Attention Logits | Value State Norms |
> |:--------:|:----------------:|:-----------------:|
> | 1-layer  |       19.2       |        1.3        |
> | 3-layer  |       21.1       |        0.7        |
>
> The results show that residual state peaks strengthen the extreme-token phenomena in three-layer transformers compared with 1-layer transformers.

---

> ### Comment · Reviewer_JE77 · 2024-11-26
> **Thank you for the response and further comparison**
>
> I am happy with the overall response and have decided to raise my score.

---

### Author Response · Authors · 2024-11-22

We sincerely thank the reviewers for their valuable comments and for recognizing our contributions to the theory and understanding of LLMs. Below, we address some general points.


## Theoretical Contributions
While most reviews focus on the practical implications of our work, we believe our work is particularly impactful for the theory community.
1. Identifying the minimal conditions that give rise to extreme-token phenomena is essential for advancing our understanding of them. From this perspective, the BB model serves as a foundational sandbox for studying these phenomena, as it reproduces extreme-token behaviors that quantitatively align with those observed in real LLMs.
2. Our simplification of the loss function introduces novel techniques for analyzing complex models, opening new avenues in learning theory.

## Extreme-token phenomena are statistically neutral with respect to model performance
The extreme-token phenomena represent the active-dormant mechanism, which is necessary for LLMs, therefore, rendering it neither inherently beneficial nor harmful to model performance. Our study employs models with ReLU activation and SGD optimization to investigate the mechanism underlying this phenomenon, rather than promoting its practical application.

---

### Meta-Review · Area_Chair_9NJd · 2024-12-20

**Metareview:**

This paper investigated the phenomena called "extreme-token phenomena" observed in transformer-based large language models that include attention sinks, value-state drains, and residual-state peaks. Based on a toy task with a small model, the authors identify an active-dormant mechanism. They develped theoretical analysis and empirical verification on both toy model and open-sourced pretrained LLMs.

Reviewers appreciate the valuable insights into the extreme-token phenomena and praise their theoretical and empirical analysis. However, there are common concerns on the simplified assumptions in the BB task, the limited robustness of the findings in real-world tasks, and the lack of practical benefits in the mitigating those phenomena.

All reviewers remain at borderline on assessing the contribution of this study at its current form after rebuttal. It is not sufficient for publication at ICLR yet. I would encourage authors to further develop their methodology to fill the gap between the simplified setup and the real-world application setting.

**Additional Comments On Reviewer Discussion:**

Authors share concerns on practical implication of findings from a simplified setup on the BB task, small network, and single head. While authors' rebuttal addressed many concerns, some reviewer remains unconvinced how the conclusion extends to more complicated practical settings. All reviewers remain at borderline after rebuttal.

---

### Decision · Program_Chairs · 2025-01-22

Reject